# Interference haptic stimulation and consistent quantitative tactility in transparent electrotactile screen with pressure-sensitive transistors

Kyeonghee Lim [ID][1,2,8], Jakyoung Lee[1,2,8], Sumin Kim [ID][1,2,8], Myoungjae Oh[1,2,8], Chin Su Koh[3], Hunkyu Seo [ID][1,2], Yeon-Mi Hong[1,2], Won Gi Chung [ID][1,2], Jiuk Jang[1,2], Jung Ah Lim [ID][4,5,6], Hyun Ho Jung [ID][3] ✉ & Jang-Ung Park [ID][1,2,3,4,7] ✉

Integrating tactile feedback through haptic interfaces enhances experiences in virtual and augmented reality. However, electrotactile systems, which stimulate mechanoreceptors directly, often yield inconsistent tactile results due to variations in pressure between the device and the finger. In this study, we present the integration of a transparent electrotactile screen with pressure-sensitive transistors, ensuring highly consistent quantitative haptic sensations. These transistors effectively calibrate tactile variations caused by touch pressure. Additionally, we explore remote-distance tactile stimulations achieved through the interference of electromagnetic waves. We validated tactile perception using somatosensory evoked potentials, monitoring the somatosensory cortex response. Our haptic screen can stimulate diverse electrotactile sensations and demonstrate various tactile patterns, including Morse code and Braille, when integrated with portable smart devices, delivering a more immersive experience. Furthermore, interference of electric fields allows haptic stimulation to facilitate diverse stimulus positioning at lower current densities, extending the reach beyond direct contact with electrodes of our screen.

The realization of artificial tactile sensations in electronic devices along with audiovisual technology can provide a more realistic user experience[1–5]. Historically, haptic systems have been employed in applications such as Braille for individuals with visual impairments and vibration feedback for notifications. Currently, there is a shift towards their utilization of virtual reality (VR), augmented reality (AR), and

providing feedback with display information[6,7]. For example, the integration of haptic functionality in automotive displays is increasingly imperative to enhance user interaction and safe driving experience. This transition needs a haptic system to accommodate high resolution, low power consumption, high transmittance, and the demands of large-area automotive infotainment and portable smart

[1]Department of Materials Science and Engineering, Yonsei University, Seoul 03722, Republic of Korea. [2]Center for Nanomedicine, Institute for Basic Science (IBS), Yonsei University, Seoul 03722, Republic of Korea. [3]Department of Neurosurgery, Yonsei University College of Medicine, Seoul 03722, Republic of Korea. [4]Yonsei-KIST Convergence Research Institute, Seoul 03722, Republic of Korea. [5]Soft Hybrid Materials Center, Korea Institute of Science and Technology (KIST), Seoul 02792, Republic of Korea. [6]Division of Nanoscience and Technology, KIST School, University of Science and Technology (UST), Seoul 02792, Republic of Korea. [7]Graduate Program of Nano Biomedical Engineering (NanoBME), Advanced Science Institute, Yonsei University, Seoul 03722, Republic of Korea. [8]These authors contributed equally: Kyeonghee Lim, Jakyoung Lee, Sumin Kim, Myoungjae Oh. ✉e-mail: junghh@yuhs.ac; jang-ung@yonsei.ac.kr

devices[8]. In contrast to vision, hearing, and smell, replicating the sense of touch is considerably more challenging due to the need for a sophisticated stimulation technique that offers fast-response[9], high-resolution[10], and extensive coverage[11]. This artificial sense of touch can be mainly implemented in two ways: vibrotactile and electrotactile.

A vibrotactile system transmits direct mechanical stimulation to the skin via the vibration[12,13]. A vibrotactile actuator implements tactile sensations of different intensities by controlling the vibration intensity; however, it is difficult to realize various tactile textures due to the deficiency of an adjustable parameter. Moreover, a vibrotactile system requires a high-power supply and bulky equipment, which hinders the implementation of precise tactile sensation and integration with a transparent screen of electronic devices[14–16]. In contrast, an electrotactile system uses electrical stimulation to innervate the mechanoreceptors in the skin, which the brain perceives as touch/tactile sensation. This system is able to produce a wider range of tactile textures by controlling two electrical parameters: density and frequency of the current[17,18]. The current density determines the intensity of tactile sensation by stimulating mechanoreceptors to regulate the firing numbers of the receptor potential. The frequency is related to the type of activated mechanoreceptor, which enables it to mimic more diverse tactile sensations than the vibrotactile system[19]. In addition, the electrode can encompass a range of materials, provided they possess conductivity, thereby enabling the development of electrotactile actuators with diverse color and transparency properties[20]. This capability facilitates the creation of a haptic system that is responsive to the display of not only smart portable devices but also automobiles.

However, due to the inherently high impedance of transparent electrodes (e.g. indium tin oxide and metallic nanowires) used in commercial displays, these conventional transparent electrodes were difficult to achieve a safe level of tactile-evoked electrical stimulation[21,22]. Moreover, within the realm of tactile feedback, including electrotactile, assessments have predominantly relied on qualitative evaluations derived from subjective perceptions obtained by survey-like experiments. These subjective evaluations lack quantitative criteria, posing challenges in achieving consistent tactile data collection[23]. Additionally, alterations in the skin-electrode interface due to external factors, especially touch pressure, lead to impedance variations at the interface, hindering the consistent implementation of electrotactile sensation. An increase in pressure between the electrode and the skin results in changes in contact area and distance, leading to heightened current flow even with the same applied voltage. As a result, even the same stimulation bias can yield different tactile sensations due to variations[24] in touch pressure each time[25–28].

To overcome these limitations, in this study, we report a transparent electrotactile screen that can stimulate the tactile sense of human fingertips, by integrating pressure-sensitive transistors as pressure sensors and attaching it on the display of a portable smart device. The key advantages of our haptic device, transparent pressure-calibratable interference electrotactile actuator (TPIEA), are as follows (1) Firstly, we modified the electrode surface by electroplating Platinum (Pt) nanoclusters on transparent Indium tin oxide (ITO) to increase the contact area and reduce impedance while maintaining high transmittance (>89% in visible spectrum). (2) Secondly, we used the somatosensory evoked potential (SEP) to quantitatively assess the tactile perception. SEPs measure the electrical activity of the brain in response to tactile stimulation, providing an objective and quantitative evaluation of perceived tactile sensation. (3) Thirdly, we integrated pressure-sensitive transistors (as pressure sensors) with our electrotactile screen to calibrate individual pressure variations when the finger is placed on the screen. This calibration ensured uniform tactile sensation by quantitatively monitoring the touch pressure from different fingers. (4) Finally, taking one step further, we demonstrated an unconventional application of interference haptic stimulation to our electrotactile device. Interference haptic stimulation exploits the phenomenon of interference using multiple high-frequency electric fields with offset frequencies to stimulate distant locations far from the electrode position through the generated interference waves[29,30]. This interference method allows tactile stimulation at spacing between multiple electrodes, thereby facilitating changes in stimulation position with high-resolution and remote-distance stimulation, compared to previous single-stimulation methods[31,32]. It also reduces susceptibility to external factors such as pressure. Furthermore, this interference stimulation offers the advantage of enabling stimulation at lower current densities than the single stimulation case used only two electrodes, the anode and the cathode.

By manipulating the frequency and current density of the stimulation bias, our TPIEA is able to generate diverse tactile sensations. For example, we demonstrated Morse code, Braille, and image-based tactile patterns to show the capacity of the TPIEA to deliver tactile information beyond basic tactile representation. With cognitive experiments associated with these patterns, we achieved a substantial level of tactile information transmission. The fine resolution of these tactile sensations was validated by a two-point discrimination (TPD) test, which assesses the minimum discernible distance between two points on the skin as separate stimuli[33,34]. By utilizing the electrotactile sensation and high transparency of this device, our study demonstrates the potential for extending tactile implementation to displays.

## Results

### Transparent pressure-calibratable interference electrotactile actuator (TPIEA)

The principle of electrotactile stimulation on human skin is described in Supplementary Note 1, including Supplementary Fig. 1 and Supplementary Table 1. Figure 1a illustrates the configuration of TPIEA for uniform tactile sensation by regulating finger pressure. This was achieved by integrating two components, an electrotactile actuator and a pressure-sensitive transistor (as a pressure sensor), into a single device using a layered integration. The top layer houses the electrotactile actuator, which directly interfaces with the finger skin to provide tactile feedback, while the bottom layer accommodates the pressure sensor for measuring the finger pressure.

The electrotactile actuator is circular with a diameter of 500 μm, forming a 10 × 10 array. The distance between the two active electrodes was designed to be 2 mm to satisfy the TPD threshold for fingertip sensation. To enable the use of this electrotactile actuator in transparent displays, ITO was sputtered at 100 nm as the material for the electrodes and patterned by photolithography on one side of a colorless polyimide (PI) film. Then, Pt nanoclusters were electroplated on the ITO surface to reduce the impedance of electrodes, as shown in scanning electron micrographs (SEM) of Fig. 1b. The inset shows the formation of nanoclusters on ITO. To minimize the transparency degradation caused by the electroplating Pt nanoclusters, we optimized the electroplating time to 10 s (Supplementary Fig. 2), because electroplating processes longer than 10 s can reduce the transmittance of the electrodes to below 70%. All the interconnections were covered with a photo-patternable SU-8 epoxy layer to avoid unnecessary contact. Every material for the electrotactile actuator is biocompatible for blocking the inflammatory responses on skin[35–37].

We employed a pressure-sensitive field-effect transistor (FET) as a pressure sensor due to its superior capabilities in detecting and calibrating inter-individual pressure variations. Notably, the FET's potential for extension to high-dimensional devices offers significant advantages for integration with screen devices. A comparative analysis of our sensor's performance with previously reported motion sensors is provided in Supplementary Table 2. The pressure sensor works based on the FET which consists of a channel, source, drain, dielectric layer, and gate (Supplementary Fig. 3). The top-gate of this FET was photo-patterned on the reverse side of PI film likewise using ITO. This double-side film was assembled with the bottom source/drain of FET.

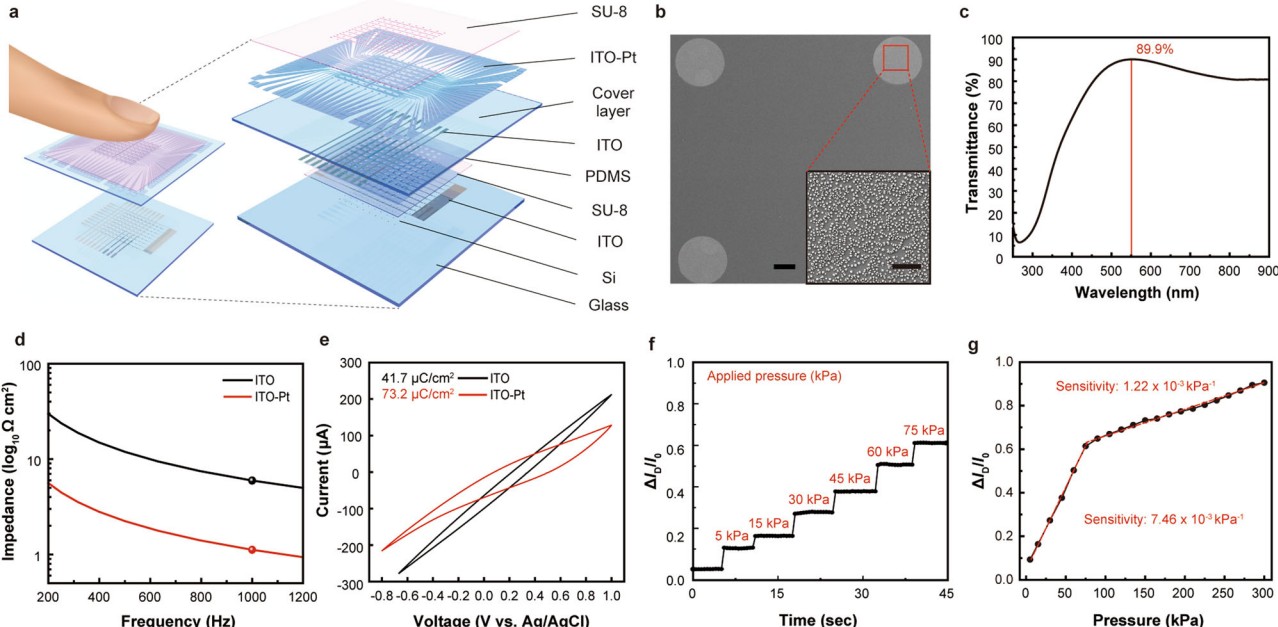

**Fig. 1 | Transparent pressure-calibratable interference electrotactile actuator.**
**a** Schematic layouts, and structural designs of the device composed of an electrotactile actuator array and pressure-sensitive transistor array. The device directly contacts the skin surface of the finger (left). **b** SEM image showing stimulating electrode and a reference electrode on the electrotactile actuator. Scale bar, 200 μm. Magnified SEM image showing the Pt nanocluster on stimulating electrode. Scale bar, 8 μm (inset). **c** Optical transmittance spectrum of the transparent pressure controllable electrotactile device range from 250 nm to 950 nm wavelength. **d** Impedance graph comparing ITO-Pt and pristine ITO over the frequency range from 200 Hz to 1200 Hz **e** Cyclic voltammetry graph recorded at a sweep rate of 100 mV s⁻¹. **f** Real-time detection of relative change in drain current of the pressure-sensitive transistor of TPIEA at pressure ranges from 5 kPa to 75 kPa. **g** Relative changes in drain current of the pressure-sensitive transistor of TPIEA by the applied pressure at the ranges from 5 kPa to 300 kPa, and the pressure sensitivity of pressure sensor (red line).

The detailed fabrication procedure of this pressure-sensitive FET was described in Methods. The device showed a high transmittance of 89.9% in Fig. 1c (excluding the substrate transmittance by using this as a baseline), which can be suitable for application in display panels (Supplementary Fig. 4). Mechanical properties such as tensile strength, flexural strength, and impact resistance of TPIEA were also examined to verify the stability of the device under repetitive use. Since the majority of the device's volume or area is occupied by the substrate, these mechanical properties are strongly influenced by the materials of the substrate (Supplementary Note 2).

The impedance was evaluated by comparing the electrode with Pt-coated ITO (ITO-Pt) and pristine ITO (with no Pt coating). The relatively large surface area of these rough Pt nanoclusters reduces the impedance of the electrode, allowing for more effective electrical stimulation at low current density[38]. At 1 kHz frequency, for example, the ITO-Pt electrode exhibited an impedance of $1.11\,\Omega\,cm^2$ (red dot) while the pristine ITO electrode presented $5.95\,\Omega\,cm^2$ (black dot), about 5 times lower value (Fig. 1d and Supplementary Fig. 5). Even after applying a 70 min electrical stimulation, the ITO-Pt electrode shows an impedance value approximately four times lower than that of the pristine ITO electrode (Supplementary Fig. 6). In addition, the charge storage capacity (CSC), which represents the electrical capacity of an electrode for electrical stimulation, was characterized by cyclic voltammetry at a scan rate of 100 mV s⁻¹ (Fig. 1e). The ITO-Pt electrode had CSC of 73.2 μC cm⁻², which was more suitable for the stimulation than the pristine ITO electrode case (41.7 μC cm⁻²). Furthermore, to examine the sensitivity of pressure-sensitive FET, real-time detection of the change in drain current was conducted during applying the external compressive pressure from 5 kPa to 300 kPa (Fig. 1f and Supplementary Fig. 7). The sensitivity was calculated as $7.46 \times 10^{-3}$ kPa⁻¹ within the pressure range from 1 to 75 kPa, and $1.22 \times 10^{-3}$ kPa⁻¹ within the pressure range from 75 to 300 kPa (Fig. 1g). The sensitivity was enough to control the human finger pressure whose range was estimated about from 30 to 200 kPa[39,40]. Furthermore, the relative changes in the drain current of pressure-sensitive FET remained consistent, with the decrease being less than 1%, over 1000 cycles of compressive pressure loadings at 200 kPa (Supplementary Fig. 8). During this pressure loading, the response time and recovery time were examined to be 21 ms and 23 ms, respectively (Supplementary Fig. 9). This FET was also evaluated by transfer and output characterization (Supplementary Fig. 10), and a calculated mobility of its channel (Si) was 111 cm² V⁻¹s⁻¹. The on/off ratio ($I_{on/off}$) and the threshold voltage ($V_{th}$) of FET were calculated as $2 \times 10^3$ and 37.6 V, respectively. Moreover, the uniformity of 100 transistors was analyzed to present Gaussian distribution profiles of $I_{on/off}$ and $V_{th}$ (Supplementary Fig. 11).

In addition to these electrical performances, the biocompatibility and safety of the TPIEA material were rigorously assessed. Various calculations, including adherence to international general standards, skin compatibility, and cell viability tests, confirmed its suitability for potential biomedical applications (Supplementary Notes 3 and 4).

## Classification of tactile sensation using SEP analysis

Before implementing tactile sensation using the TPIEA, we quantitatively evaluated how various electrical tactile stimuli are perceived in the sensory cortex by monitoring SEPs. The SEP has been used to monitor neurophysiological responses to electrical stimuli applied to the central nervous system (Supplementary Note 5 and Supplementary Fig. 12). In our experiment, the stimulation electrodes were attached to the right index finger at intervals of 2 cm, and the recording electrode was placed on the scalp to detect brain cortical signals. Figure 2a shows the short-latency SEPs measured according to the positions of this stimulation, recording, and reference electrodes. The recording electrode was located at C3, which is responsible for the right somatosensory response, and the reference electrode was placed at Fz, which

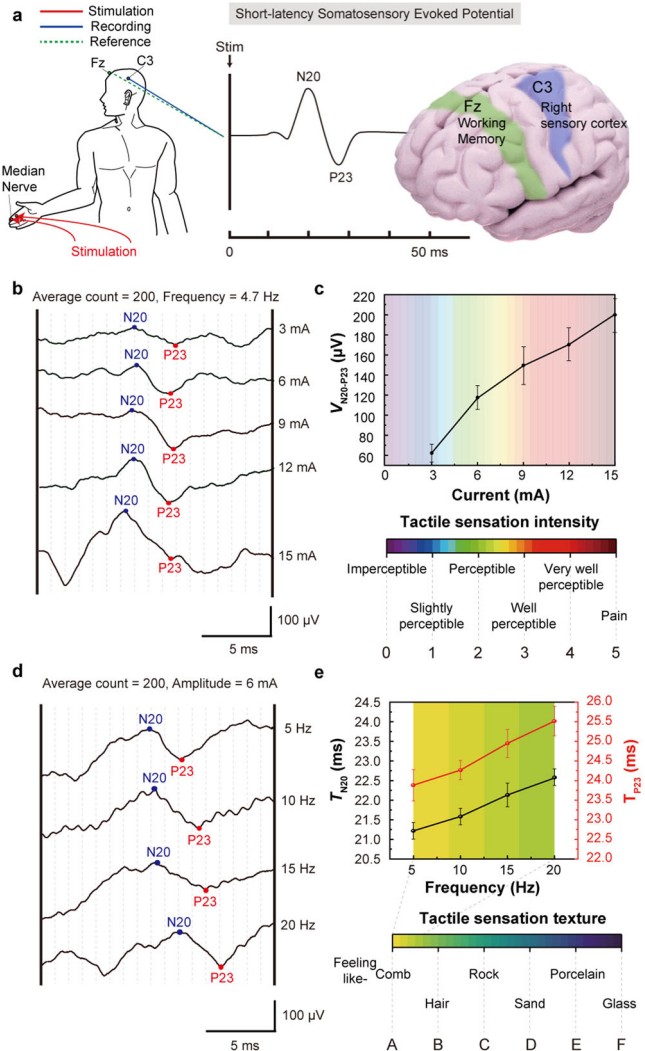

**Fig. 2 | Quantitative classification of electrotactile via SEP. a** Schematic image showing the positioning of electrodes and the identification of peaks during short-latency SEPs elicited by median nerve stimulation. **b** SEPs ranging from 13 ms to 30 ms with amplitudes of 3, 6, 9, 12, and 15 mA. **c** $V_{N20\text{-}P23}$ graph when changing amplitude and classification of the electrotactile intensity, along with the categorization of electrotactile intensity, which is divided into six levels from 0 mA to 15 mA denoted by the numbers 0–5. **d** SEPs ranging from 13 ms to 30 ms at frequencies of 5, 10, 15, and 20 Hz **e,** $T_{N20}$ and $T_{P23}$ graph when altering the frequency from 5 Hz to 20 Hz, along with the classification of electrotactile texture, which is divided into six levels from 5 Hz to 200 Hz represented by the letters A – F. Data in c and e are presented as mean ± SEM.

is responsible for immediate and sustained attention[41] (Supplementary Fig. 13 and 14). To measure the changes in the electrotactile sensation perception, the post-stimulus peak time and peak potential difference of N20 and P23 were used as indicators of the SEP. The SEP was evaluated on three subjects, and each SEP was measured with 200 averages. When the amplitude and frequency of the stimulus were modulated, both indicators changed.

First, we measured the changes with the increase in the intensity of electrotactile sensation. The subjects' tactile perception was collected by gradually increasing the amplitude from 1 mA while fixing the frequency of the current at 4.7 Hz. The tactile perception of the subjects changed from "Imperceptible" to "Slightly perceptible" at approximately 3 mA, to "Perceptible" at 6 mA, to "Well perceptible" at 9 mA, and to "Very well perceptible" at 12 mA. When the stimulus intensity was increased to 15 mA, the subjects complained of "Pain".

We categorized the "Tactile sensation intensity (tactile intensity)" by dividing the perceived intensity into six levels, from "Imperceptible" to "Pain". Therefore, SEPs were collected when a current corresponding to each tactile intensity was applied (Fig. 2b). The peak potential difference between N20 and P23 ($V_{N20\text{-}P23}$) increased to 62.33 μV at 3 mA, 118.27 μV at 6 mA, 149.92 μV at 9 mA, 167.24 μV at 12 mA, and 201.75 μV at 15 mA as the amplitude increased, confirming that different electrotactile sensations were perceived in the brain. Since this tactile intensity is also used in subsequent experiments, for convenience, each level corresponded to a number and color from 0 to 5 (Fig. 2c). In contrast to the peak potential, the post-stimulus peak time of N20 ($T_{N20}$) and P23 ($T_{P23}$) did not change significantly with the applying stimulus intensity ($T_{N20}$ = 21.5–20 ms, $T_{P23}$ = 23.25–23.9 ms) (Supplementary Fig. 15).

Next, to measure changes in tactile perception dependent on stimulus frequency, the tactile intensity was fixed at the "Perceptible" level (stimulus current amplitude = 6 mA), and the frequency of electrical stimulation was varied from 5 Hz to 200 Hz. The subjects reported a smoother sensation with increasing frequency, categorizing the "Tactile sensation texture (tactile texture)" as "Comb" in around 5 Hz, "Hair" around 40 Hz, "Rock" around 80 Hz, "Sand" at 120 Hz, "Porcelain" at 160 Hz, and "Glass" at 200 Hz. The SEP peak was measured up to 20 Hz to ensure that the measured electrophysiological signals were not influenced by frequency variation. When measuring SEP signals, $T_{N20}$ was delayed from 21.24 ms at 5 Hz to 22.53 ms at 20 Hz, and the $T_{P23}$ was delayed from 23.89 ms at 5 Hz to 25.11 ms at 20 Hz. Since this tactile texture was also used in subsequent experiments, for convenience, each level was assigned a letter and color from A to F (Figs. 2d, e). In contrast, $V_{N20\text{-}P23}$ showed no significant trend, with inconsistent values of 100 μV, 110 μV, 75 μV, and 80 μV measured as the frequency changed. (Supplementary Fig. 16).

This qualitative analysis of the electrotactile sensations and corresponding SEP signals collected from the subjects, when the frequency and amplitude of the stimulus current were adjusted, provided evidence that these electrotactile sensations were processed differently in the brain. A higher amplitude induced a higher tactile intensity and increasing the frequency changed the tactile type from a tapping sensation to a smoother sensation. This suggests that electrical parameters such as amplitude and frequency could affect electrotactile sensation perception in the brain and could lead to the implementation of various sensations when these parameters are adjusted.

## The pressure effect on tactile actuation

Our electrotactile system implemented tactile sensation by adjusting the amplitude and frequency of the stimulation current, and SEP analysis quantitatively confirmed that the electrotactile sensation was perceived differently in the brain. Here, this electrotactile system can generate different tactile sensations depending on the touch pressure. When a finger simply touches a certain object, the mechanoreceptors under the epidermis are weakly stimulated. Mechanoreceptors activated by weak stimulation deliver the receptor potentials to the central nervous system at a few repetitions, therefore, the human perceives this sensation as a weak sensation. On the other hand, a stronger force from the finger makes the human perceive tactile sensation as relatively strong. This difference in stimulation intensity results from the variability of the contact area and distance between the skin and object. This same phenomenon occurs in the electrotactile system. Under the same stimulus condition, if the touch pressure between the skin and the electrode is weakened, the contact area narrows and the distance between the mechanoreceptors under the epidermis increases, resulting in a weak sensation. Conversely, when the pressure increases, the contact area becomes wider and the distance to the mechanoreceptors becomes closer, resulting in a stronger sensation. Since the skin-electrode impedance changes with the pressure,

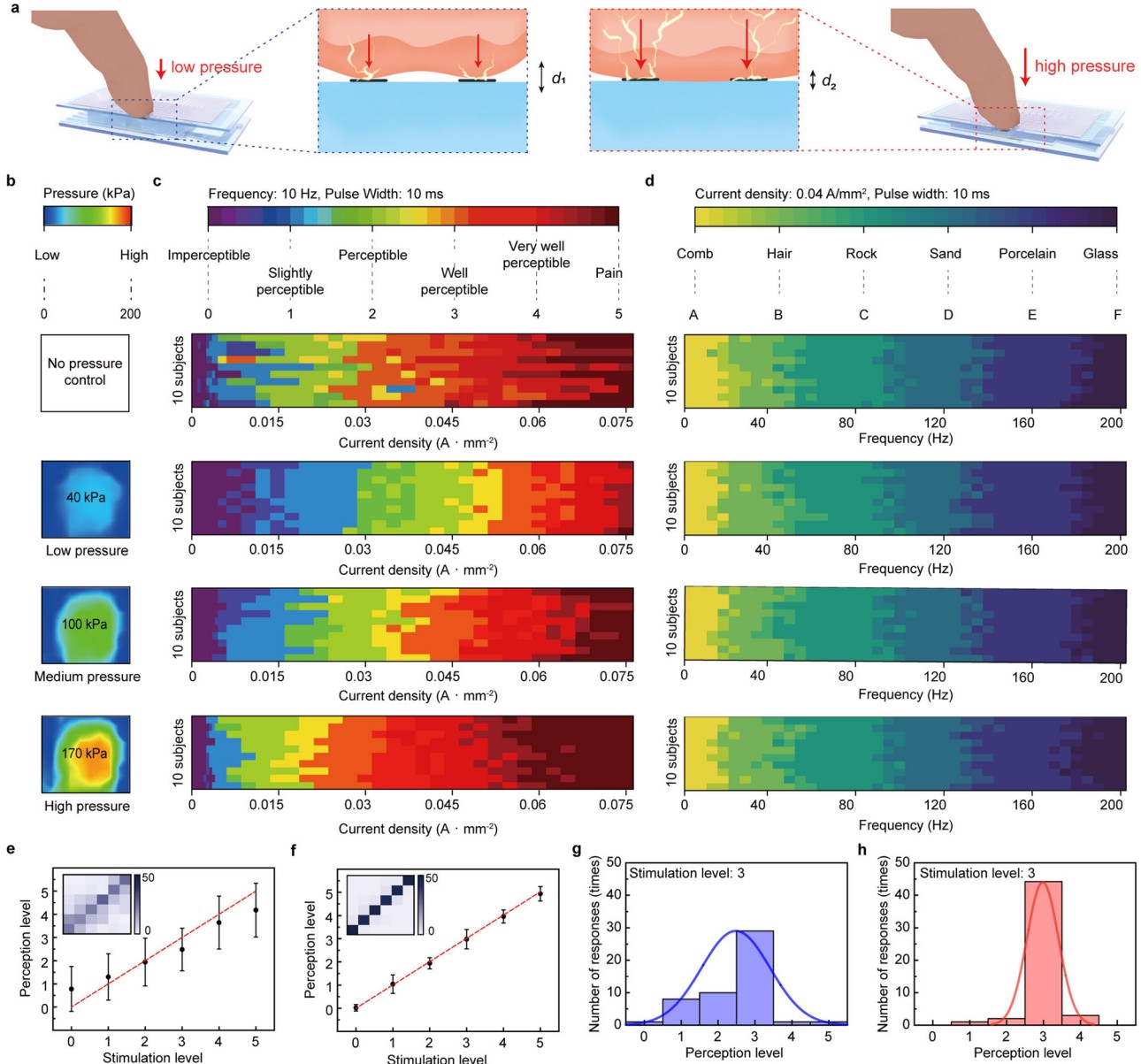

**Fig. 3 | Experimental result of tactile perception test with pressure control.**
**a** Schematic illustrations of the operating mechanism for pressure sensing and electrotactile stimulation applied on the finger at low pressure(left), and at high pressure (right). **b**, **c**, **d** Color maps recording the electrotactile sensation responses of 10 subjects when the current density and frequency are varied under four pressure control conditions. **b** The four pressure conditions were: uncontrolled pressure, low pressure (<40 kPa), medium pressure (-100 kPa), and high pressure (>170 kPa). **c** Electrotactile intensity for each finger pressure when the current density was changed from 0 to 0.075 A mm⁻². **d** Electrotactile texture for each finger pressure when the frequency is changed from 0 to 200 Hz. Perception response to the stimulation composed of six levels of current amplitude and confusion matrix (inset) with uncontrolled pressure (**e**) and with 100 kPa of controlled pressure (**f**). Data are presented as mean ± SEM. Distribution of responses at 3 of stimulation levels with uncontrolled pressure (**g**), and with a controlled pressure of 100 kPa (**h**).

electrotactile systems, which are based on electrical stimulation, are also affected.

To standardize this variability, we monitored the applied pressure to control it at a certain value when subjects placed their fingers on our TPIEA. Pressure sensing involved varying the thickness of an air-dielectric layer of the pressure-sensitive FET located on the lower layer of this TPIEA. We used a polydimethylsiloxane (PDMS) elastomer which contained air holes. The vertical contraction of PDMS increased the capacitance between the gate and channel, which led to an increase in drain current, and the relative change in drain current was used to measure the touch pressure (Supplementary Fig. 17). Figure 3a illustrates that a finger is pressing the TPIEA with the touch force. This FET can monitor variations in touch pressure to calibrate the electrotactile

sensation (Supplementary Note 6). To verify the correlation between finger pressure and electrotactile sensation, a perception test was examined under various finger pressure conditions when the stimulation parameters were changed. The four pressure conditions are as follows: (1) uncontrolled pressure, (2) low pressure (40 kPa), (3) medium pressure (100 kPa), and (4) high pressure (170 kPa) (Fig. 3b). Under each pressure condition, we examined how the electrotactile sensation changed when the stimulation current density and frequency were varied in ten subjects. First, we varied the current density flowing through the skin at a frequency of 10 Hz and a pulse width of 10 ms. Figure 3c shows the tactile intensity of the responses for each condition. The criteria for tactile intensity were the same as the tags and colors established by SEP analysis. In uncontrolled pressure condition

(no FET operation), the inter-individual difference in touch pressure was relatively large, resulting in large inter-individual variation in perception. Even for the same current, the participants showed differences in their perception, with some reporting a lower tactile intensity with the increasing current density. However, when the finger pressure of all subjects was controlled to a specific pressure of 40, 100, and 170 kPa, there was less significant inter-subject variation. Furthermore, when the same current density was applied, the higher the finger pressure, the higher the tactile intensity. For example, when stimulated with a current density of 0.03 A mm$^{-2}$, the tactile intensity at a finger pressure of 40 kPa was about level 2, but at 100 kPa and 170 kPa it was level 3 and 4, respectively. This is consistent with the mechanism that higher pressures result in more current flowing, which allows for the perception of higher tactile intensity. When the current density of the stimulating current was higher than 0.075 A mm$^{-2}$ (the current density at which most subjects had the highest perceptual level), subjects also felt warmth at their fingertips (Supplementary Note 7).

Next, we investigated how varying the frequency of the current changed the texture of tactile sensation across the four pressure conditions (Fig. 3d). Here, the current density of the stimulus was applied to be 0.04 A mm$^{-2}$ and the pulse width was 10 ms. The criteria for tactile texture were the same as those for tag and color established in the SEP analysis. When stimulation was applied at high frequencies close to 200 Hz, the perceived level approached a glass-like tactile sensation, but this linear trend was independent of pressure and occurred in all pressure conditions. The pulse width was also independent of pressure. Pulse width denotes the length of time for which the electrical stimulus is active, delineated into periods of activation (on time) and deactivation (off time) (Supplementary Fig. 18). By modifying the pulse width of electrotactile, it is possible to elicit two significant alterations in tactile perception. Initially, an increase in pulse width enables the generation of a sustained pressure and an elongated pulling sensation, as the stimulation can be consistently applied, in contrast to shorter pulse widths (Supplementary Fig. 19). Secondly, manipulation of the duty cycle can serve as a method for individuals to differentiate between 0 and 1, facilitating binary information transmission. It was subsequently verified through Morse code within the application.

This result indicates that the effect of pressure on the realization of electrotactile sensation is limited to tactile intensity and not tactile texture. Therefore, the current density should be adjusted to compensate for the pressure-dependent tactile inconsistency. To determine the impact of adjusting finger pressure on perception accuracy, we compared the accuracy in the perception of electrotactile stimulation with and without this pressure control. A randomly arranged stimulation consisting of six levels of tactile intensity (from level 0 to level 5) was applied for ten blindfolded subjects using TPIEA with a pulse width of 10 ms and a frequency of 10 Hz. This procedure was repeated five times for each subject. In the absence of pressure control, the finger pressures applied by each of the ten subjects were different respectively, resulting in a wide distribution of responses to each stimulation level. (Fig. 3e). The accuracy was low at 50%, which can be examined visually by a confusion matrix as shown in the inset. This result indicates that the variable pressures produce a different sensation from the electrotactile actuator even if the electrical parameters of the applied current are constant, therefore, it is important to standardize the variability of the finger pressure. Then, we repeated the same examination while controlling the finger pressures from all subjects at 100 kPa, the typical pressure of a normal object manipulation by humans[34]. Consequently, the perceptual accuracy was increased to about 93% (Fig. 3f, Supplementary Fig. 20). Perception responses at finger pressures of 40 kPa and 170 kPa are also presented in Supplementary Fig. 21. The distribution of perception at stimulation level 3 is widely spread before pressure control, whereas the

distribution is narrowly concentrated at controlled pressure (Figs. 3g, h and Supplementary Fig. 22). Based on these results, all subsequent experiments were pressure-calibrated and performed at safe current densities, which were confirmed by additional experiments to distinguish between pain and touch sensation. In particular, distinguishing between pain and touch sensations is essential to safely implementing tactile sensations, so additional Visual Analog Scale and SEP experiments were conducted and compared again to determine the precise distinction between pain and sensation, as well as the stage of pain (Supplementary Note 8). This meticulous calibration ensured the reliability and accuracy of the experimental outcomes. In addition, the skin-electrode impedance changes with not only the pressure but also the humidity of skin, it is important to control the humidity for consistent results. Therefore, another factor that affects electrotactile sensation, skin humidity, was also controlled and adjusted to maintain consistent conditions throughout the tests under varying skin moisture conditions (Supplementary Note 9).

## Various tactile patterns implemented using TPIEA

To evaluate the precision of the tactile sensation provided by TPIEA, the TPD test was conducted. Precise implementation of electrotactile sensation is essential because the density distribution of mechanoreceptors is approximately one order of magnitude higher at the fingertip, compared to other body parts (e.g. forearms and toes)[42]. The TPD test was employed to determine whether TPIEA satisfied the TPD threshold and delivered high-resolution tactile sensation to the human fingertip (Supplementary Note 10). Three different distances between the two stimulation points were selected for this test: 2, 2.83, and 4 mm. These distances were adjusted by modifying the combination of the active electrodes, as detailed in Supplementary Fig. 23. The results indicated that 78.3% of subjects were able to discern a separation of 2 mm, 90% for 2.83 mm, and 98.3% for 4 mm on their fingertips. The reason for these variations is attributed not to the device's performance but rather to difference in the distribution of mechanoreceptor cells among individuals, leading to differences in TPD thresholds. Additionally, our experimental findings align with previous research on TPD test that the threshold for tactile detection in the human fingertip begins at a minimum of ~ 2 mm and escalates to approximately 5 mm[43,44]. Therefore, this outcome proves that the TPIEA is capable of generating diverse patterns with high-resolution tactile sensation, in accordance with the TPD. In addition, the capability of repetitive use was verified through a nerve fatigue test involving 30 trials per person. The results showed no significant change in the intensity of electrotactile sensation (Supplementary Fig. 24).

The TPIEA contains an array of 10 × 10 electrotactile actuators to produce a range of haptic sensations by adjusting the current density, frequency, and pulse width of the actuators. Localized and broad stimulation patterns can be generated by combining stimulation locations, indicating the potential for transmitting information by TPIEA. By manipulating the stimulus parameters using a single actuator, temporal patterns of tactile sensation, such as morse code, can be achieved. Additionally, spatial patterns, such as Braille, can be created by determining the stimulus area and adjusting the position of the stimulus point. By combining temporal and spatial patterns, feedback on the realized tactile pattern can be obtained from the subject. Subjects were instructed to place their fingers on the TPIEA to experience an electrotactile sensation and then provide feedback based on the tactile information (Supplementary Fig. 25). Initially, morse code, which is a communication method representing text characters as specific sequences of two different signal durations (dots and dashes), was employed. Dots were represented by a pulse width of 10 ms, while dashes were represented by a pulse width of 500 ms. To ensure clear distinction of each code by the subjects, the frequency was set to 1 Hz, and the current density was 0.04 A mm$^{-2}$ for all conditions. Morse code enables the representation of alphabets

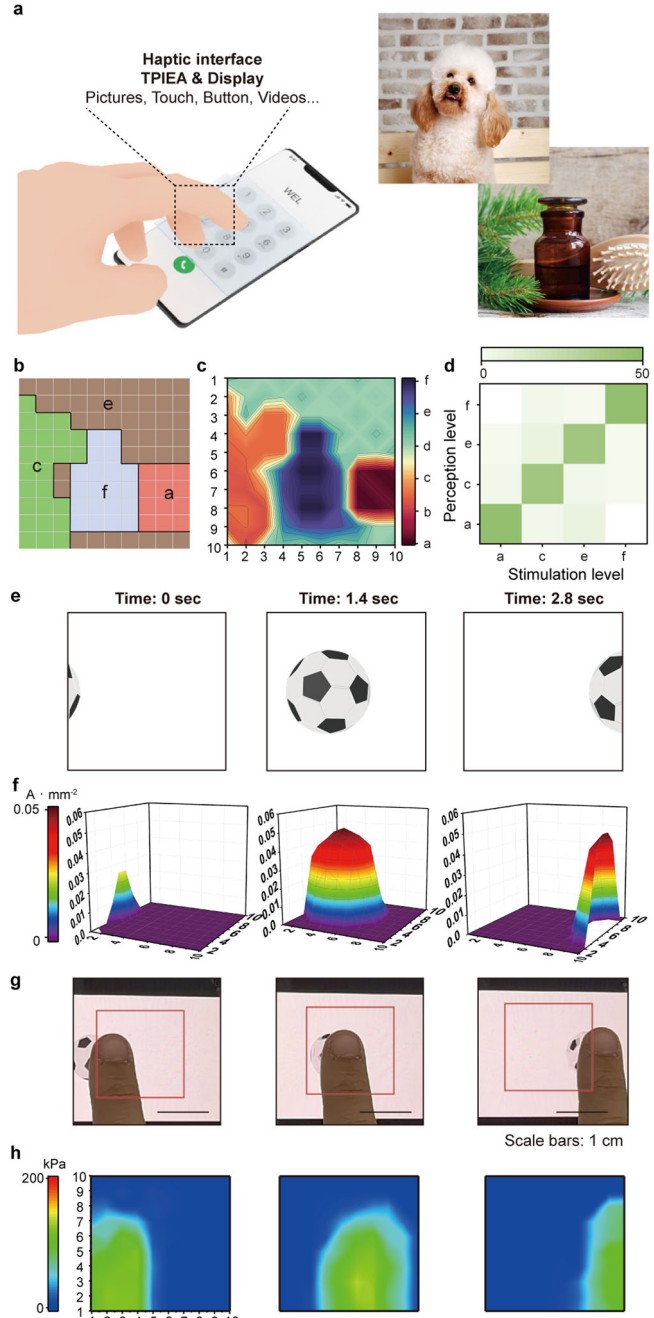

**Fig. 4 | Implementation of virtual tactile sensation corresponding to the display. a** Schematic illustration of TPIEA applied on the display panel. The device can transmit tactile information corresponding to the images displayed (Image source: Gettyimages). **b** Schematic image of a 10 × 10 TPIEA array divided for tactile mimicking corresponding to picture. **c** Color maps depicting the electrotactile sensation to be implemented according to the divided zones. **d** Confusion matrix result of 50 perception test trials corresponding to each electrotactile sensation. **e** Images of the rolling ball captured from video (Video source: Septadi, Adobe Stock). **f** Spatiotemporal color maps of the current density of tactile stimulation corresponding to ball movement in 3D. **g** Finger movements that follow the direction in which the ball is rolling due to the parallel change in electrotactile sensation implemented. **h**, Spatiotemporal color maps of finger pressure distribution applied to the display of an electronic device during finger movement.

using a combination of these two signals. When blindfolded subjects were randomly stimulated 100 times to distinguish morse code, the average accuracy was 96%, and all alphabets could be distinguished with high accuracy as depicted in the confusion matrix

(Supplementary Fig. 26 and 27). When asked to transcribe a simple sentence based on the morse code implemented at their fingertips while not blindfolded, the subjects were able to write sentence.

Subsequently, we demonstrated that Braille, a tactile writing system developed for individuals with visual impairments, can effectively convey spatial tactile information. Accurate reading of Braille necessitates the spatial differentiation of the stimulated part of the Braille cell. This system comprises six raised dots arranged in 2 × 3 matrices, known as a Braille cell, with the arrangement and number of dots distinguishing one character from another. To ensure precise information transmission, 2 × 3 stimulating electrodes were chosen at 2 mm intervals. The positioning and sequence of stimuli were organized to communicate the alphabetical information such as "L", "O", "V", and "E" (Supplementary Fig. 28a). Stimulation was administered at a frequency of 10 Hz, a pulse width of 10 ms, and a current density of 0.04 A mm$^{-2}$. The frequency was adjusted to 10 Hz to achieve spatially accurate stimulation, in contrast to Morse code, which is designed to discern the order of each code. Finite element analysis (FEA) simulation using the COMSOL Multiphysics program demonstrated that the stimulating electrodes at 2 mm intervals do not interfere with each other (Supplementary Fig. 28b). This analysis provided mapping data of current flow when electrotactile sensation of Braille pattern was applied to multiple electrodes and simulated all patterned alphabets (Supplementary Fig. 29 and 30). During blindfolded experiments, subjects were presented with random stimuli implementing Braille alphabets using the same approach as in the Morse code, resulting in the collection of 100 feedback responses. The average accuracy, expressed as a confusion matrix (Supplementary Fig. 28c and Supplementary Fig. 31), was 83.6%. When the Braille experiment was repeated at 4 mm intervals, the accuracy significantly increased to 95.1% (Supplementary Fig. 32 and 33). By implementing the two representative methods of information transmission through tactile sensation, morse code and Braille, TPIEA can demonstrate sufficient information transmission capabilities similar to existing tactile communication systems. It has also shown that adjusting the parameters and locations of electrical stimulation, it can expand the stimulation patterns from a single actuator to multiple actuators, thereby achieving various electrotactile sensations (Supplementary Video 1).

## Display-integrated electrotactile sensation

Upon analyzing the human test results for various tactile implementations using TPIEA (incorporating pressure calibration), we mapped the types of sensations experienced when adjusting current density and frequency (Supplementary Fig. 34). This mapping allows for flexible parameter adjustments in electrical stimulation to obtain desired tactile sensations. Furthermore, TPIEA can be seamlessly integrated atop displays without compromising image and video quality, due to its high transmittance in the visible light range.

To verify the applicability with a display, images with various tactile information were selected to confirm whether corresponding tactile sensations could be implemented. In Fig. 4a, the top image depicts a puppy with fur seated on a wooden chair against a rock background. The bottom image displays, from left to right, a plant with grass, a glass bottle, and a comb against a rock wall background. These two images contain a total of 6 types of tactile sensations corresponding to comb (a), fur (b), grass (c), wood (d), rock (e), and glass (f). Then, as shown in Fig. 4b and Supplementary Fig. 35, the 10 × 10 actuator array of TPIEA was divided to correspond to the images, and each region was assigned a tactile sensation to be implemented by varying current frequency and density locally. The current density for comb, fur, grass, wood, rock, and glass was set to 0.03, 0.025, 0.02, 0.05, 0.035, and 0.03 A mm$^{-2}$, respectively, and the frequency was specified as 10, 40, 120, 160, 80, 200 Hz. Figure 4c shows the mapped images of these electrotactile sensations to be implemented by TPIEA at each location. To evaluate the accuracy of

these sensations, perception experiments were conducted five times with 10 blindfolded subjects. The subjects could accurately distinguish between the different sensations in the first image at a rate of 88.6% and in the second image at a rate of 88% as shown in Fig. 4d (Supplementary Fig. 36).

TPIEA is not limited to static images; it can also be applied to moving videos. By adopting a video of a soccer ball rolling through the TPIEA screen, electrotactile sensations were implemented to follow the movement of this ball. The position and intensity of the electrical stimulation varied depending on the ball's movement. To mimic the ball's shape, current was applied in 10 steps from the center to the outer edge, ranging from 0.005 to 0.05 A mm$^{-2}$ with 10 Hz. The ball rolled from left to right at a constant speed for 3 s (Fig. 4e), and the stimulation current's position and density were adjusted accordingly. This pattern was displayed through 3D mapping to demonstrate how the electrical stimulations were applied (Fig. 4f). A blindfolded subject was instructed to place a finger on the screen's left side and follow the detected electrotactile sensation, matching the movement of this ball (Fig. 4g). The finger's pressure, measured using a pressure-sensitive FET, showed consistency at approximately 100 kPa (Fig. 4h). In this way, TPIEA effectively conveyed tactile information corresponding to visual images shown through a display, projecting various tactile information in pictures and videos (Supplementary Fig. 37 and Supplementary Video 2).

Moreover, the TPIEA can be fabricated on different substrates, making it adaptable for use in flexible display technologies. We have also implemented a flexible version of TPIEA using a PI substrate, demonstrating its adaptability (Supplementary Fig. 38). This versatility in substrate indicates that TPIEA can be integrated into a wide range of display applications, enhancing its potential for future technological innovations[45].

## Interface haptic stimulation

The electrotactile system induces artificial tactile sensations by stimulating mechanoreceptors using electrical impulses. Hence, through modification of the electrical stimulation technique, it is feasible to produce a novel tactile perception distinct from the current electrical touch sensation. Notably, the incorporation of interference stimulation into the electrotactile system enhances the effectiveness of tactile sensations by stimulating the mechanoreceptors in a novel manner, rather than simply adjusting electrical stimulation parameters. Interference haptic stimulation exploits the interference between two high-frequency electric fields with offset frequencies to achieve electrotactile stimulation at a deep skin position beyond the electrode surface through the generation of interference waves (Supplementary Note 11). The superimposition of two electric fields with frequencies above 1 kHz produces an envelope wave with an offset frequency at a position determined by the sum of the vectors of these two electric fields. Consequently, interference stimulation can be applied based on the principle that electrical stimulation in the frequency band above 1 kHz does not elicit a response from neurons, even if current flows, due to the intrinsic low-pass filtering phenomenon[46]. To discern the difference when applying interference stimuli to the fingertip compared to a single stimulus (with no interference), electrical potential signals were measured on the skin surface away from the stimulation point (Fig. 5a). Figure 5b plots the electrical potential signals associated with both single stimulation and interference stimulation. The single electrical stimulation was recorded at frequencies of 10 Hz, 50 Hz, 100 Hz, and 1000 Hz. In interference stimulation, one pair of electrodes remained constant at 100 Hz, while the other pair delivered currents corresponding to 1000 Hz, 1001 Hz, and 1010 Hz. The current density of each electrode was 0.015 A mm$^{-2}$ for single stimulation and 0.01 A mm$^{-2}$ for interference stimulation. From 10 Hz to 100 Hz, a pulse wave waveform with an average peak-to-peak of 10.72 mV, 9.45 mV, and 6.57 mV was observed. However, at 1000 Hz, the signal

current density significantly decreased to 1.63 mV. In the interference stimulation, when the second current was superimposed at 1000 Hz, no envelope wave appeared. Yet, when the offset frequency was given at 1001 Hz and 1010 Hz, an envelope wave was measured. The electric field interference produced a minimum peak-to-peak amplitude of 1.262 mV and a maximum amplitude of 4.86 mV. Despite the interference stimulus having half the current density of the single value, a comparable electrical potential signal was acquired as a result of this interference. The frequency of each envelop wave was 1 Hz and 10 Hz, consistent with the offset frequency value of each electric field, confirming the application of interference stimulation to the finger skin.

To determine recognition when an interference stimulus is presented, a recognition test was conducted compared to a single stimulus, as shown in Fig. 5c. The recognition test was repeated randomly with 10 subjects, 10 times for each stimulus method. The single stimulus showed a high recognition of 100% for 10, 50, and 100 Hz, excluding 99% at 200 Hz. However, no one recognized it at 1 kHz. Also, no tactile sensation was recognized under unoptimized interference stimulation (1000 Hz + 1000 Hz), a condition where an interference stimulus was applied but no envelope wave appeared. In contrast, the stimulations with offset frequencies of 1010 Hz, 1050 Hz, and 1100 Hz showed an average recognition of 99.6%. This confirms that electrotactile sensation is not realized in bands above 1 kHz, and electrotactile sensation is selectively realized under optimized interference conditions with offset frequencies and not under all interference stimuli (Supplementary Video 3).

To confirm electrotactile sensation through interference stimulation, a tactile perception test was conducted with changes in current density and frequency, similar to the single stimulation. The finger pressure was kept at 100 kPa, and the stimulation current was applied equally to both pairs of electrodes. In interference stimulation, intensity tended to increase as the stimulation current was increased (Fig. 5d). Similarly, as the offset frequency changed, a similar pattern of tactile textures emerged as before (Fig. 5e). Since interference stimulation utilizes superimposed currents generated from interference, it was possible to create tactile sensation at a lower current than the single stimulation case in terms of current applied to the electrode. When comparing the threshold current densities at which we started to perceive the "Perceptible" level as the frequency changed, on average, the value for single stimulation was 0.0119 A mm$^{-2}$, and the value for interference stimulation was 0.00736 A mm$^{-2}$. This is a 38.15% lower current density than that of the single stimulation, indicating better stimulation efficiency using interference than single stimulation (Fig. 5f). Furthermore, the interference stimulation was less sensitive to changes in pressure than single stimulation because it was a deep stimulation due to the sum of the vectors of the electric field rather than a surface stimulation. In 10 subjects, we assessed variations in electrotactile intensity of the interference stimulation across a range of current density from 0.0175 A mm$^{-2}$ to 0.0225 A mm$^{-2}$ in response to changes in pressure, in comparison to a single electro-static stimulus. When the touch pressure was not calibrated, the tactile intensity of interference stimulation was limited to the "Perceptible" to "Well perceptible" level, whereas the single stimulation showed a larger variation from "Imperceptible" to "Very well perceptible". Overall, there was a slight increase in the intensity of perception with increasing pressure, but it was less sensitive to pressure than single stimulation (Fig. 5g). We measured the threshold current for perceptible levels when each finger pressure was at low, medium, and high-pressure levels (Fig. 5h). The threshold current density for single stimulation and interference stimulation entering the "Perceptible" level (at each finger pressure) was compared. For single stimulation, the threshold current density decreased by 79.96% from 0.0257 A mm$^{-2}$ to 0.00515 A mm$^{-2}$ when the pressure increased from low to high. For interference stimulation, the threshold current decreased by 56.33% from 0.01225 A mm$^{-2}$ to 0.00535 A mm$^{-2}$.

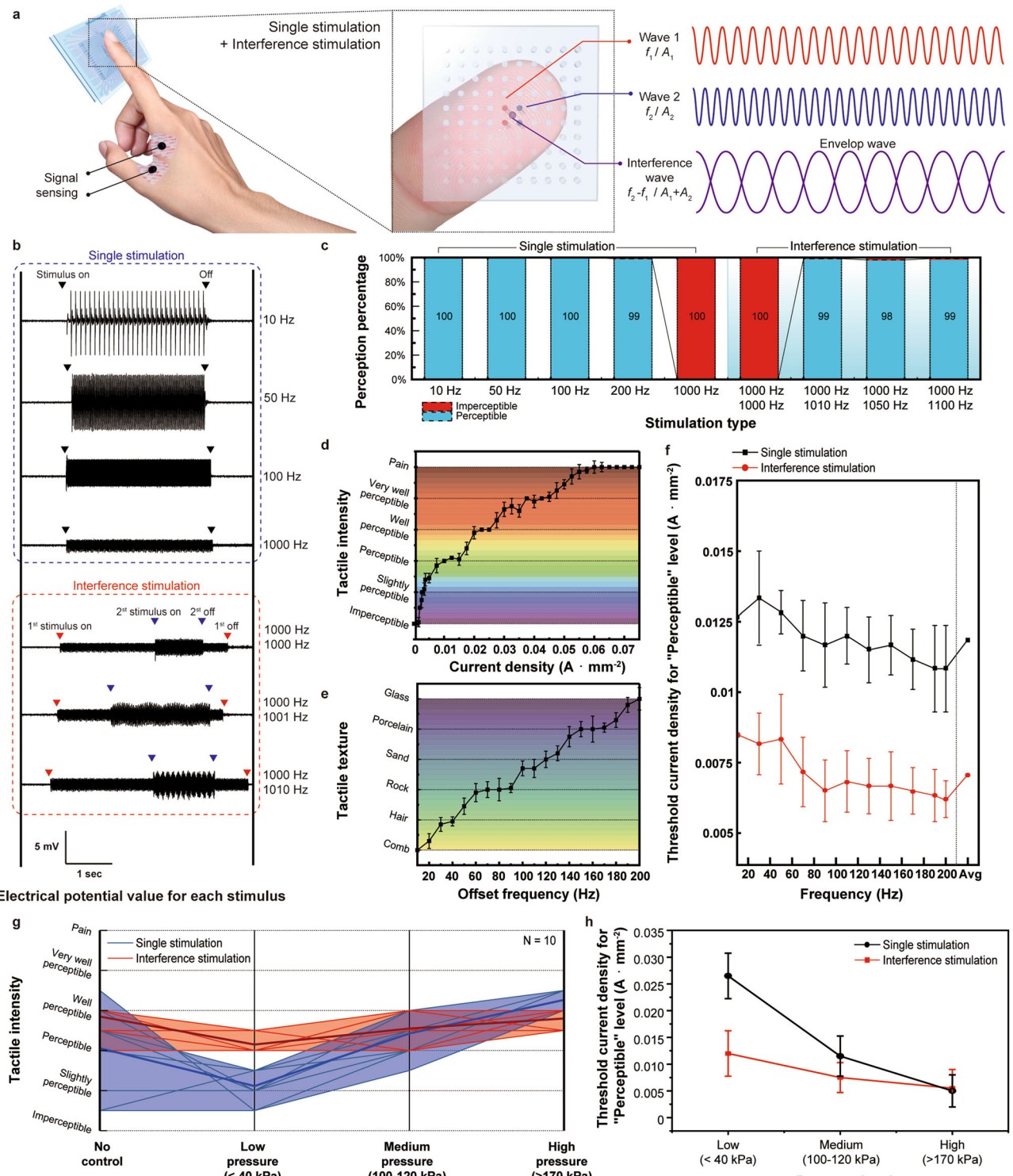

**Fig. 5 | Electrotactile haptic system with interference stimulation. a** Schematic of interference stimulation in electrotactile system. An Ag/Cl patch is affixed above the skin surface to measure the electrical potential differences. **b** Electrical potential signals measured based on the type of electrotactile stimulation. **c** Perception test corresponding to the type and frequency of electrotactile stimulation. **d** Variations in the tactile intensity when the current density adjusted. **e** Variations in the tactile texture when the offset frequency modified. **f** Threshold current density required to reach the "Perceptible" level when altering the frequency of the single stimulus and the offset frequency of the interference stimulation. **g** Precepted tactile intensity level for single stimulation and interference stimulation as a function of the finger pressure level. **h** Different threshold current density required to reach the "Perceptible" level at each pressure level. Data in d, e and f are presented as mean ± SEM.

Compared to single stimulation, the rate of change in current density for achieving the same tactile sensation with interference stimulation was approximately 66.42% lower, indicating that it was less sensitive to pressure compared to single stimulation. Therefore, incorporating interference into an electrotactile system enables the attainment of electrotactile sensations that are less influenced by touch pressure, as it stimulates deep skin areas, and can be also achieved at lower current densities.

## Enhancing tactile sensations: integrating interference with single stimulation

Interference haptic stimulation enables the creation of electrotactile sensations at the center of four active electrodes. When combined with single stimulation, this method enables diverse stimulus positioning, allowing the application of electrotactile stimuli even in areas without electrodes. This versatility allows for the implementation of stimulation at various locations using the same geometry of electrode arrays, facilitating the complex tactile realization of display image correspondence. The stimulation positions that can be created for each stimulation pattern in the finger area are illustrated in Fig. 6a and Supplementary Fig. 39. As in previous studies, the "Normal site" can be generated by wrapping four cathodes around one anode electrode, and adjusting the position of the four cathodes allows for a "High-resolution normal site" (Supplementary Fig. 40) in a single stimulation[10]. Additionally, a "Center site" can be obtained by placing one cathode and one anode in parallel to stimulate the center of each electrode. The introduction of interference stimulation further extends the stimulation sites to the center of these four electrodes ("Interference site") (Supplementary Fig. 41). The superiority of spatial resolution and low current density of the

interference stimulation was demonstrated in Supplementary Fig. 42 and Supplementary Table 3.

The combination of single stimulation and interference stimulation enables the generation of high-resolution patterns. The precise patterns were chosen to represent the ancient four elements: flame, droplet, pinwheel, and leaf. As presented in Fig. 6b, creating an image of a flame requires not only the stimulation sites obtained with single stimulation but also interference sites (Supplementary Fig. 43). All four stimulus locations were utilized to create a complex curved section, and the stimulus locations and conditions are shown in Fig. 6c. To achieve a consistent tactile intensity, a current density of 0.0175 A mm$^{-2}$ was used for the single stimulus, and 0.01 A mm$^{-2}$ for the interference stimulus.

The ability to implement these intricate patterns was made feasible by the interference stimulation, which enables the use of stimuli with greater resolution than previously achievable. While the number of stimulation sites with a single stimulation was 118 when limited to the finger area, the introduction of interference stimulation increased it to 130, resulting in a total of 505 stimulation sites across the entire area, equivalent to 155.86 stimulation sites per 1 cm$^2$, representing an increase in resolution of approximately 32.08% (Fig. 6d). Ten blindfolded subjects were presented with a random pattern ten times for each elemental pattern and asked to identify the presented pattern. The results exhibited an accuracy as high as 92.5%, indicating that complex image tactile patterns can be realized by introducing a corresponding interference stimulus to the display (Fig. 6e and Supplementary Fig. 44). This haptic stimulation also demonstrated the capability to differentiate patterns with similar configurations, such as fire patterns with slightly different orientations or numbers of flames (Supplementary Fig. 45 and 46). The high-resolution of stimulation provided by TPIEA, along with its capacity to modulate a diverse range of tactile sensation, enables its application in various settings, such as integrating visual and tactile components with VR equipment (Supplementary Note 12 and Supplementary Fig. 47).

## Discussion

In summary, our study presents a transparent, pressure-calibratable interference electrotactile actuator that adeptly replicates diverse and consistent tactile sensations. Our findings show that electrotactile stimulations generated by this device can be recognized and classified in the sensory cortex of the brain, as confirmed by quantitative SEP analysis and qualitative cognitive testing. This device demonstrates the ability to simulate a spectrum of tactile sensations through meticulous adjustments in frequency, current density, and pulse width of the stimulating current. To address a common limitation in previous electrotactile systems, we integrated pressure-sensitive transistors in the electrotactile actuators, minimizing the impact of varying finger pressure on the stimulated mechanoreceptors. The integration of this transistor array ensures the consistency of tactile feedback by precisely monitoring finger pressure.

Through cognitive experiments involving Morse code and Braille, we demonstrated the device's capability to selectively stimulate different positions, providing diverse electrotactile sensations and suggesting broader applications beyond the specific tasks tested. Additionally, the device's high transparency allows it to effectively mimic textures from images and videos displayed on the screen. Moreover, our exploration into interference stimulation reveals promising avenues with various advantages. Introducing interference stimulation allows for haptic stimulation with enhanced resolution and depth compared to the single stimulation. This interference method facilitates diverse stimulus positioning, extending the application of electrotactile stimuli even to areas without electrodes, enabling stimulation at various locations using a consistent array structure of electrodes. This interference between multiple electric fields also allows electrotactile stimulation at a deep skin position beyond the

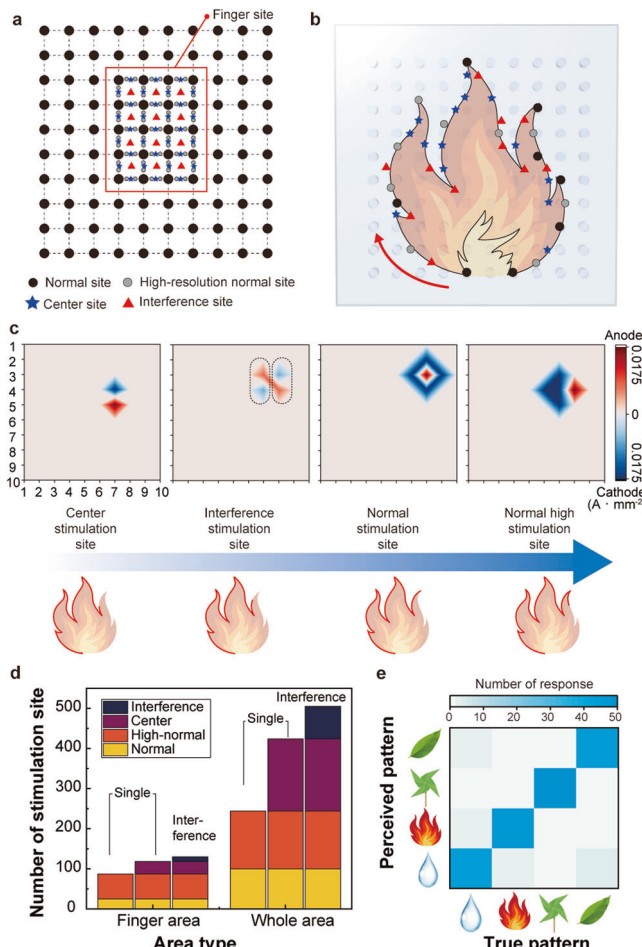

**Fig. 6 | High-resolution electrotactile pattern enabled through interference stimulation. a** Schematic illustration of the distribution of stimulation sites for various stimulation types. The red squares indicate the area covered by the finger. **b** Schematic image of the complex pattern and stimulation sequence of a flame created by combining single stimulation sites with interference stimulation sites. **c** Stimulation patterns that are combined to create a pattern of current density mapping data. **d** The total number of stimulation sites in the finger area and the total area. **e** Confusion matrix results of 50 perception trials corresponding to each pattern.

electrode surfaces, effectively reducing sensitivity to touch pressure. Another advantage is the potential for interference stimulation at lower current densities, due to the amplification achieved through the superposition of waves. This paves the way for further advancements in haptic displays, including VR, AR, and automotive displays.

## Methods

### Fabrication of device

In terms of a pressure sensor fabrication, the pressure sensor is assembled structure of channel-source/drain substrate, dielectric layer, and top gate electrodes. First, for the fabrication of channel-source/drain substrate, Si channel (thickness, 205 nm; channel length, 20 μm; channel width, 200 μm) was transferred to a SU-8 2002 (MicroChem) coated (spin coating, 4000 rpm/40 s) glass substrate from a silicon-on-insulator wafer (buried oxide, 400 nm) using a PDMS stamp. Source/drain electrodes (thickness, 100 nm) were patterned by photolithography after ITO sputtering deposition. The passivation layer, which was photolithographically patterned by negative photoresist, SU-8 2002 (thickness, 1 μm) covered the electrodes excepted to the channel region. Secondly, for the formation of air-dielectric (thickness, 50 μm), polydimethylsiloxane (PDMS) was cast in a Si-mold. PDMS was made by mixing base and curing agent (10 wt%) using a planetary centrifugal mixer (THINKY Corp., Japan). After spin coating (1500 rpm/30 s) of PDMS on Si mold, it was cured for 3 h at 70 °C. Si mold has an array of vacant holes to give each FET side wall. Lastly, the gate electrodes (thickness, 100 μm) were fabricated onto the PI film (thickness, 25 μm) by lift-off lithography of ITO in a similar fabrication of source/drain electrodes. On the opposite side of PI film, electro-tactile actuator is fabricated. LOR 3 A (spin coating, 4000 rpm / 40 s, baking 185 °C/6 min) (KAYAKU, USA) and S1818 (spin coating, 3000 rpm/30 s, baking 115 °C/3 min) (Dow Chemical, USA) was patterned by negative photoresist. After developing, electrotactile electrodes (thickness, 100 nm) were patterned by lift-off lithography after ITO sputtering deposition. The passivation layer, which was photolithographically patterned by negative photoresist, SU-8 2002 (thickness, 1 μm) covered the electrodes excepted to the stimulating actuator region. All materials used in the experiments are summarized in Supplementary Table 4.

### Electrodeposition of platinum nanoclusters

The electrodeposition of platinum nanoclusters involved the preparation of a 100 mL electroplating solution, consisting of 100 mL of DI water, 20 mg of lead acetate trihydrate (Sigma-Aldrich), and 1 g of platinum tetrachloride (Sigma-Aldrich), which was mixed at room temperature. The solution was then subjected to ultrasonic vibration for 20 min and filtered to eliminate impurities. Electroplating was performed by transferring ions between the cathode (ITO-Pt electrode) and the anode (a Pt electrode), both of which were immersed in the electroplating solution and connected to a SourceMeter (2400 Series SMU, Keithley). The electroplating reaction took place under an electrical voltage of 5 V and a compliance current of 3 mA for an optimized duration.

### Impedance and charge storage capacity spectroscopy

The pristine ITO and ITO-Pt electrodes were assessed for impedance and charge storage capacity spectroscopy in a DPBS solution (Sigma-Aldrich). Impedance measurements were carried out using a multi-channel potentiostat (PMC-1000, AMETEK) within a frequency range of 0.01 to 100 kHz. Charge storage capacity was determined at a rate of 100 mV s$^{-1}$ and the area was computed through graph analysis. Capacitance was determined at a frequency of 1000 Hz with 25 data points. To assess electrical fatigue, electrical stimulation was administered at a current density of 0.01 A mm$^{-2}$, a frequency of 10 Hz, and a pulse width of 10 ms, followed by the measurement of electrode impedance within a 1 kHz bandwidth. The impedance was assessed

three times, with stimulation durations of 5, 10, 20, 30, 40, 50, 60, and 70 min.

### Characterization of pressure-sensitive transistor

The electrical performances of pressure-sensitive FET, including transfer and output characteristics, were characterized using a probe station (Keithley 4200-SCS). The transfer characterization was examined at a drain voltage of 1 V. The output characterization was evaluated with the gate voltage ranging from 20 to 70 V in 10 V increments. To assess the pressure-sensing performance, the pressure was applied to the sensor using a high-precision motorized z-stage (Mark-10 ESM303).

### Cell viability test

We cultured SH-SY5Y cell line, which is a human neuroblastoma cell line, on the device and applied electrical stimulation with a current intensity of 0.015 A mm$^{-2}$, a frequency of 10 Hz, and a pulse width of 10 ms for 0 min (control group), 5 min, 30 min, 60 min, and 180 min. As a result of Calcein AM, fluorescence images confirmed with an optical microscope exhibit consistent trends for each group.

### Evaluation of skin safety in mice during long-term electrical stimulation

The experiment with mouse was performed in compliance with all the ethical regulations under a protocol that was approved by the Institutional Animal Care and Use Committee of Yonsei University (application no. IACUC-A-202405-1851-01). C57BL/6 male mice (6/-8 weeks old) were purchased from Orient Bio (Seoul, Korea). The animals were maintained on a 12/12 h light/dark cycle in a temperature-controlled room (23 ± 2 °C). The skin of each group of mice was removed the hair with an electric razor and hair removal gel. The mouse skin tissue was cut (0.8 cm × 0.8 cm) and fixed in 4% paraformaldehyde for 24 h at 4 °C after stimulation. Stimulation was administered for 6 min, comprising 5 min of electrical stimulation and 1 minute of rest, with experiments conducted under varying durations (no stimulation, 1 set − 6 min, 5 sets − 30 min, and 10 sets − 60 min). The stimulation parameters remained consistent at 10 Hz frequency, 10 ms pulse width, and 0.015 A mm$^{-2}$. They were embedded in paraffin and sectioned at thickness of 5 μm. Following deparaffinization and rehydration, sections were stained with an H&E stain kit according to the protocol provided by Vector (Brussels, Belgium). The image of the section was obtained under a microscope (Olympus BX53M).

### Experiments with human subjects

The experiments with human subjects were performed in compliance with all the ethical regulations under a protocol that was approved by the Yonsei University Institutional Review Board (application no. 7001988-202303-HR-1809-02) via a haptic interface with no additional human-subject risk, following the provided study guidelines. The subjects in all experiments were individuals of both genders, between the ages of 22 and 40. Given that display and device usage are not restricted by gender or sex, we included both male and female subjects in our recruitment process. All subjects provided written informed consent about the experimental procedure. All subjects were trained to manipulate the electrotactile system with the help of experimenters until they were familiarized with the sensation of electrical stimulation. Subjects were asked to clean their fingers with alcohol to minimize effects of moisture before the test, maintaining a 36–42% skin humidity. For all tests, subjects underwent a 5− to 10 min familiarization phase to acclimatize themselves with the task. All of the subjects prepared for dull sensations by having a 5 min break time after conducting a 10 min cognitive experiment to prevent the reduction in tactile perception. Also, even when they restarted, they sanitized their hands with alcohol and wiped them off to maintain a 36−42% humidity level on the skin surface. A temperature and humidity control unit was

utilized to maintain a consistent level of humidity for controlled environment: 26 °C and under 43.1% humidity. Also, all cognitive experiments were conducted under the advice and observation of the neurosurgeon. The authors affirm that human research participants provided informed consent for publication of the images in Supplementary Figs. 14 and 47, and Movies 1, 2 and 3.

## Experiment 1: two-point threshold test

Twenty healthy adult subjects (age, 28–35 years; female individuals) performed the tests for two-point discrimination and accuracy measurements via a haptic interface with no additional human-subject risk, following the provided study guidelines. All the subjects provided consent before the test. Two-point discrimination was confirmed by dividing the stimulation intervals into 2 mm, 2.83 mm, and 4 mm with. The process was repeated three times for each subject.

## Experiment 2: SEP experiments with human-subject

Three healthy adult subjects (age, 25–30 years; male individuals) were arranged in the test. The subjects had no previous or current neurological/motor deficits. Nihon Kohden's Neuromaster G1 MEE-2000 equipment was used, the setting sensitivity is 20 μV, the pulse interval is 2.0 ms, the average count is fixed at 200, and the potential is recorded for 50 ms after stimulation was applied. The stimulation electrode is situated on the right index finger, the reference electrode is placed at Fz on the scalp, and the recording electrode is located at C3 on the scalp. The cortical regions associated with the reference and recording electrodes in relation to SEPs are the working memory and right sensory cortex, respectively, with the targeted SEP peaks being N20 and P23. All electrodes' impedance was below 20 kΩ. Electrical stimulation was applied while varying the current and amplitude.

## Experiment 3: tactile perception at different current densities, pulse width and frequency

Ten healthy adult subjects (age, 22–35 years; six males and four females) were arranged in the test to test the relationship between the perceived tactile intensity and stimulation. First, the stimulation signal consisted of an AM-modulated 10 Hz pulse wave that lasted 10 ms. The current density of the modulated wave was gradually increased until the subject reported a noticeable sensation. It was subsequently increased until the subject started to feel uncomfortable, and the reading was regarded as "Pain" level. Second, the stimulation signal consisted of an frequency-modulated $0.04\,A\,mm^{-2}$ pulse wave that lasted 10 ms. After increasing the frequency to 200 Hz, the tactile sensation distribution was recorded. In order to investigate the changes in electrotactile sensation according to pulse width, we varied the pulse width to 10 ms, 100 ms, 300 ms, 500 ms, 700 ms, and 900 ms at a frequency of 1 Hz and a current density of $0.015\,A\,mm^{-2}$, and asked the subjects to respond with "Brief touch," "Pulling sensation," "Sustained pressure," or "Very long stimulus." The pulse width test was repeated five times for each subject. Also, five healthy adult subjects (aged 25–30; three males and two females) responded to variations in electrotactile sensation due to changes in current density and changes in frequency, using TPIEA on a flexible substrate.

## Experiment 4: human-subject perception test controlling pressure

Ten healthy adult subjects (age, 22–35 years; six males and four females) were arranged in the test to test the relationship between the perceived tactile intensity and stimulation. The amplitude of electrical stimulation was subsequently increased until the subject started to feel uncomfortable, and the reading was regarded as "Pain" level. Also, the frequency of electrical stimulation is changed from 10 Hz to 200 Hz pulse wave that lasted 10 ms. The electrical tactile sensation distribution was recorded when each electrical stimulation parameter is changed, during pressure control. The pressure was controlled in 3

conditions (low pressure of 40 kPa, medium pressure of 100 kPa, and high pressure of 170 kPa). The result of the perception test in 3 conditions were compared to the result of the test with no pressure control. Just before subjects put their finger on the device, they cleaned their finger with a pure towel. The pressure from the finger was detected in the dark room using the FET of pressure sensor, where drain voltage of 1 V and gate voltage of 40 V. The voltage was applied by SourceMeter (2400 Series SMU, Keithley), and we monitored the drain current of FET in real-time comparing it to the initial drain current which was recorded before the test. The process was repeated five sets (from level 0 to 5) for each subject.

## Experiment 5: morse code and Braille perception haptic test

Twenty healthy adult subjects (age, 28–40 years; ten male and ten female) performed the tests for distinguishing Morse code and Braille perception with TPIEA. Subjects put the left index finger on the device while blindfolded, and then responded the perceived code and pattern. It proceeded by randomly changing the order, via a haptic interface with no additional human-subject risk, following the provided study guidelines. The process was repeated five times for each subject.

## Experiment 6: various tactile implementation test with display

Twenty healthy adult subjects (age, 28–40 years; ten male and ten female) performed the tests for distinguishing various type of electrotactile sensation. Subjects put the left index finger on the device while blindfolded. First, six types of electrotactile sensation (rock, comb, fur, wood, glass, grass) was mimicked corresponding two images. Subjects perceived tactile sensation through the left index finger on the device while blindfolded. The process was repeated twice for each subject, and an additional 10 subjects were randomly selected to be tested once more. Second, subjects put their left index finger on the end of left part of TPIEA. When the electrotactile sensation through TPIEA is applied according to the movement of the ball rolling from the left side, the finger moves according to the tactile via a haptic interface with no additional human-subject risk, following the provided study guidelines. All images and video used have been granted permission for use (Gettyimages, Adobe Stock).

## Experiment 7: electrotactile perception test with interference stimulation

Ten healthy adult subjects (age, 22–35 years; six males and four females) were arranged in the test to test the electrotactile perception test with interference stimulation. First, Ag/AgCl electrodes were affixed to the hand and linked to a multi-electrode array for electrophysiological assessment. The signal was recorded using the Neuro-Phys/NeuroSorter software (JAGA Systems, Jiang-hi) with a sampling rate of 3060 Hz and a 60 Hz notch filter during the recording process.

Second, each subject underwent a recognition experiment to ascertain the recognition of the stimulus under single stimulus conditions with varying frequencies. Similar procedures were followed for interference stimuli, distinguishing between optimized and nonoptimized conditions. This experiment was repeated 10 times for each subject.

Third, the experiment to differentiate between tactile intensity and texture during interference stimulation. The amplitude of electrical stimulation was subsequently increased until the subject started to feel uncomfortable, and the reading was regarded as "Pain" level. Also, the frequency of electrical stimulation is changed from 10 Hz to 200 Hz pulse wave that lasted 10 ms. The electrical tactile sensation distribution was recorded when each electrical stimulation parameter is changed. This experiment was repeated once for each subject. Also, to investigate pressure-dependent changes in tactile intensity, intensity was measured by applying different finger pressures at a current density of $0.0175\,A\,mm^{-2}$ during interference

stimulation. The same experiment was conducted under a current density of 0.0225 A mm$^{-2}$ for single stimulation. This experiment was repeated once for each subject.

Furthermore, to determine the threshold current density required to elicit the "Perceptible" sensation at various pressure levels, the current density was incrementally raised from 0 A mm$^{-2}$ to 0.035 A mm$^{-2}$ for each type of stimulus (single, interference) until participants reported perceiving the "Perceptible" level of sensation. The current density at that point was then documented.

Last, to discern high-resolution electrotactile patterns, subjects were blindfolded and instructed to place their fingers on the TPIEA, then move their fingers along the direction of the stimulus to learn the pattern. We combined normal stimulation sites achievable through single stimulation, encompassing normal, high-resolution, center, and interference sites resulting from disruptive stimulation. Subsequently, we depicted four distinct patterns—flame, droplet, pinwheel, and leaf. The stimulation was administered sequentially rather than concurrently, enabling the finger to trace the image along the stimulation position from the pattern's inception. The stimulation frequency (offset frequency) was consistently set at 10 Hz, while the intensity was maintained at a perceptible level. The results were presented in a confusion matrix, and the pattern recognition experiment was repeated five times in a randomized order for each subject.

### Experiment 8: high-resolution assessment on interference electrotactile sensation
Ten healthy adult subjects (age, 25–35 years; five males and five females) were arranged in the recognition test to illustrate the potential of interference haptic stimulation in discerning patterns with similar configurations through higher resolution stimulation. The initial experiment involved subjects matching the number of points in flames when the number of points in flames were adjusted. The flames consisted of 3, 4, and 5 points, with distinct arrangements enabling differentiation through high-resolution patterns, but the large form is the same. Subjects with occluded vision, entailed testing each point quantity five times in random order per subject, repeated ten times. Next, a subsequent experiment focused on matching the direction of flames. Following the preparation of flames with an identical number of points in flame, the two primary flame directions (right, left) were distinguished. They were categorized into four groups based on the orientation of the main flame points: (left, left, right) (right, right, right) (left, left, left) (left, right, right). The cognitive experiment mirrored the methodology of the preliminary study, with each subject repeating it five times.

### Experiment 9: observation of the skin state when using TPIEA
The stimulation parameters included a frequency of 10 Hz, pulse width of 10 ms, and intensity of 0.015 A mm$^{-2}$ on skin. After 1 h of stimulation, one subject (age, 27 years; one female) was instructed to compare the skin to non-stimulated skin to ensure that no skin irritation or redness has occurred.

### Experiment 10: VAS test and SEP signal analysis for pain assessment
A VAS test was administered alongside an electrotactile sensation intensity assessment to five healthy adult subjects (age, 25–30 years; three males and two females) with the finger pressure calibrated to medium (100 kPa). Following the establishment of a VAS rate of 5 electrotactile sensation corresponding to "Very well perceptible" level, we recorded the VAS responses when the current density was increased from 0.05 A mm$^{-2}$ to 0.1 A mm$^{-2}$. Also, we re-measured SEP, the baseline for electrotactile sensation, in three subjects (aged 25–30; two males and one female). Initially, SEP were obtained at 10 mA, a level at which no pain sensation is reported. Subsequently, data were gathered and analyzed for SEP ranging from 12 mA to 20 mA. The indicators utilized to correlate pain and touch sensation with SEP were the emergence, latency, and amplitude of P1-2 and N1-2 corresponding to the initial peak. This experiment was repeated once for each subject.

### Experiment 11: threshold of electrotactile sensation during long-term stimulation
Three healthy adult subjects (age, 25–30 years; two males and one female) were arranged in the test to evaluate the perception threshold with long-term stimulation. The experiment involved repeating the electrotactile intensity test 30 times at medium finger pressure and measured the threshold current density ($A$) from the "Perceptible" level to the "Well perceptible" level, as well as the threshold current density ($B$) from the "Well perceptible" level to the "Very well perceptible" level. This experiment was repeated ten times for each subject.

### Experiment 12: electrotactile sensation implement on VR
One subject (age, 27 years; one female) was instructed to VR device (Meta Quest 2; Oculus Quest) and affix the TPIEA to the dorsal surface of their hand. The texture simulation was implemented to be activated with specific parameters: a current density of 0.03 A mm$^{-2}$, a frequency of 40 Hz, and a pulse width of 10 ms. Tactile feedback is integrated into the experience by delivering the stimulus prior to virtual contact with the hand during tasks such as bubble touch within the VR environment. TPIEA aimed to enhance the immersive quality of the virtual experience by synchronizing visual and tactile stimuli, pausing the task execution upon sensation detection, and resuming the task when no tactile feedback was detected.

### Experiment 13: electrotactile sensation in a varying humidity condition
Skin humidity levels were categorized into four groups: dry (30–36%), normal (36–42%), hydrated (43–50%), and very hydrated (50–60%) states. Prior to the experiments, subjects measured the humidity levels of their skin by skin analyzer (RoHS compliant; China) and then proceeded electrotactile intensity experiment. The amplitude of electrical stimulation was subsequently increased until the subject started to feel uncomfortable, and the reading was regarded as "Pain" level. Throughout the experiment, pressure was initially standardized to a medium level of 100 kPa.

### Statistical analysis
All data were presented as mean ± standard error of the mean (SEM). Statistic calculations of $p$ value were performed using an open-source code of MATLAB.

### Reporting summary
Further information on research design is available in the Nature Portfolio Reporting Summary linked to this article.

## Data availability
The main data generated in this study are provided in the Supplementary Information/Source Data file. The raw data used in this study are available in the Figshsare database. Any additional requests for information can be directed to, and will be fulfilled by the corresponding authors. The data that support the findings of this study are available in Figshare with the identifier(s) (doi:"https://doi.org/10.6084/m9.figshare.26170447"). Source data are provided with this paper.

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

## Acknowledgements

This work was supported by the Ministry of Science & ICT (MSIT), the Ministry of Trade, Industry and Energy (MOTIE), the Ministry of Health & Welfare, and the Ministry of Food and Drug Safety of Korea through the National Research Foundation (2023R1A2C2006257), ERC Program (2022R1A5A6000846 and RS-2024-00406240), and the Korea Medical Device Development Fund grant (RMS 2022-11-1209 / KMDF RS-2022-00141392). This work is funded by Korea Institute of Science and Technology (KIST) Institutional Program (2E33191 and 2E33190). This work was also supported by Institute for Basic Science (IBS-R026-D1).

## Author contributions

K.L., J.L., S.K., and M.O. contributed to ideation, conducted the literature research. K.L. participated in the comprehensive planning of the experiment, fabrication of the TPIEA device, execution of tactile perception experiments, participant recruitment, data management, and overall manuscript composition. J.L. was responsible for fabricating and assessing the pressure sensor characteristics and conducting cognitive experiments, while H.S. was involved in measuring finger pressure. M.O. contributed to the overall interference stimulation experiment, and S. K. provided oversight, participated in cognitive experiments, contributed to the design, and offered guidance. Y.-M.H. conducted assessments of skin stability and performed animal testing in safety trials. W.G.C. participated in the measurements of SEP and statistical analysis of data. J.J. provided oversight in the pressure sensor operations. J.A.L. characterized and analyzed the mechanical properties of the device. C.S.K. conducted SEP measurements, and H.H.J. supervised the overall SEP measurements and perception experiments. J.-U.P. supervised, reviewed, and edited the manuscript. All authors participated in the writing and review of this manuscript. These authors contributed equally: Kyeonghee Lim, Jakyoung Lee, Sumin Kim, Myoungjae Oh.

## Competing interests

The authors declare no competing interests.
