## [Peer Review File · Nature Communications]

Interference haptic stimulation and consistent quantitative tactility in transparent electrotactile screen with pressure-sensitive transistorsREVIEWER COMMENTS

Reviewer #1 (Remarks to the Author):

This paper presents a transparent electrotactile screen with pressure-sensitive transistors to achieve virtual tactile display. As a prerequisite for the electrical tactile stimulation, the detection of contact pressure is crucial and the approach of this work is good. However, if we want to monitor the pressure and the tightness of the simulator, it is actually not a promising solution to use a resistive or transistor-like pressure sensor. It will inevitably increase the complexity of the circuit and crosstalk interference. There are many other way to realize the detection of the pressure and even the approaching motion [Nat. Commun. 14, 3315 (2023)]. I hope the author can further explain their logic to clarify the advantages and necessity of this pressure-sensitive FET. The decrease in robustness of the entire system is also a problem. How about the effective and reliable range of the applied pressure? Besides, there are still some unsolved issues with this article, and I believe it needs to undergo revisions before it can be accepted.

1. It is usually difficult to balance touch and pain sensation in electrotactile systems. Can data or other quantifiable strategies be proposed to address this issue?
2. Concerning the safety of the device during long time operation, is the electrical stimulation it employs completely harmless to human skin and tactile nerve? Voltage leakage during operation may result in personal injury and what measurements have been taken to prevent such occurrences?
3. Repeated stimulation can lead to a decrease in a person's perceptual ability and the long-time electrical stimulation at the same location may cause nerve paralysis.. What are the proposed solutions to mitigate or prevent this reduction in tactile perception? Has the fatigue test been conducted? The author needs to confirm whether the magnitude of the electrical stimulation current meets the general standards (for example, the GB-9706 standards for medical devices).
4. I found that the whole device is made of rigid materials. What level of tensile strength, angle of bend, and impact resistance can this device tolerate? Is it possible to construct the device to be completely flexible? (Adv. Funct. Mater. 2018, 28, 1802989.)?
5. Although this device is currently utilized solely as a sensor, as indicated, potential applications in diverse fields such as VR, AR, and automotive displays have been suggested. Can the authors elaborate on the feasibility of applying this technology across these varied domains?
6. It is necessary to provide some video materials to show the operation of the devices and the reviewer can evaluate the completeness of the function.

Reviewer #2 (Remarks to the Author):

The manuscript titled "Interference haptic stimulation and consistent quantitative tactility in transparent electrotactile screen with pressure-sensitive transistors" by Lim et al. reports a transparent pressure-calibratable interference electrotactile actuator (TPIEA). The exploration to use somatosensory evoked potential (SEP) to quantitatively assess the tactile perception and use pressure sensors to calibrate individual pressure variations is interesting

and quite effective. Moreover, interference haptic stimulation is utilized to increase the resolution and reduce the current density. Given the novelty and depth of this work, Reviewer is in support of its publication in Nature communications after major revisions addressing the following:

1. In Fig. 1g, the pressure sensitivity should be characterized under more pressure stages to ensure its accuracy. Moreover, the reason why there are two pressure sensitivities should be clarified.
2. In Fig. 3f, only 100 kPa is adopted in the examination. It's better to try more pressures to verify the importance of pressure control.
3. Besides pressure, skin wetness is also one of the important parameters that influences the electrotactile sensation. Maybe there should be some discussions here.
4. The influence of current density and frequency on electrotactile sensation has been extensively evaluated in this work. So how about pulse width? How does the pulse width affect the electrotactile sensation? Moreover, in Supplementary Fig. 16, the dots and dashes in the electrical simulation are distinguished by different pulse widths (10 ms and 500 ms). But it seems that the same 10 ms pulse width with different pulse numbers (e.g., 1 pulse and sequential 20 pulses) can achieve the same results.
5. Interference haptic stimulation is utilized to increase the resolution in this work. There is also a super-resolution strategy in "Science advances 8 (36), eabp8738". Please compare these two strategies and have a discussion.
6. In Fig. 6d, interference haptic stimulation is utilized to increase the recognition accuracy among four patterns. But these four patterns are totally different from each other. It is suggested to further evaluate the high-resolution strategy by distinguishing patterns with similar configurations such as curves with slightly different angles and ellipses with small changes in aspect ratios.
7. A suggestion to reorganize the figures. The aspect ratio of the current figures seems too large.

Reviewer #3 (Remarks to the Author):

[Response to Reviewer #1]

Comment : This paper presents a transparent electrotactile screen with pressure-sensitive transistors to achieve virtual tactile display. As a prerequisite for the electrical tactile stimulation, the detection of contact pressure is crucial and the approach of this work is good.

However, if we want to monitor the pressure and the tightness of the simulator, it is actually not a promising solution to use a resistive or transistor-like pressure sensor. It will inevitably increase the complexity of the circuit and crosstalk interference. There are many other way to realize the detection of the pressure and even the approaching motion [Nat. Commun. 14, 3315 (2023)]. I hope the author can further explain their logic to clarify the advantages and necessity of this pressure-sensitive FET. The decrease in robustness of the entire system is also a problem. How about the effective and reliable range of the applied pressure?

Response to Comment:

We appreciate the reviewer's valuable comment. As the reviewer noted, transistor-based pressure sensors have a higher complexity compared to other types, such as capacitive and piezoelectric pressure sensors. Nevertheless, we have chosen to use a field-effect transistor (FET) as a pressure sensor due to the following advantages that make it suitable for our purposes.

First, our primary purpose in utilizing the pressure sensor for our electrotactile system is to calibrate pressure variations resulting from inter-individual force differences. Pressure-sensitive FETs can detect these differences effectively and provide rapid feedback due to their fast response time and wide detectable range. We have demonstrated a reliable range of applied pressure from 5 kPa to 300 kPa, covering pressures more than twice of those required for normal object manipulation (Supplementary Fig. 7) [*Physical Therapy*, **62**, 965 (1982)]. Furthermore, the response time of our pressure sensor was examined as 21 ms (Supplementary Fig. 9), about 10 times faster than that of a capacitive pressure sensor (Supplementary Table 2).

In addition, our electrotactile system is designed for integration with screen devices. In this context, the high spatial contrast of a FET-based sensor array allows the electrotactile system to be extended to high-dimensional screen devices, making it particularly useful type in the display industry.

To highlight the advantages of pressure-sensitive FETs, we have included a table comparing the performance of various types of pressure sensors in the revised Supplementary Materials (Supplementary Table 2).

Moreover, the robustness of this entire system has been demonstrated through cyclic compressive repetition tests (Supplementary Fig. 8). The pressure sensor maintained its performance, showing consistent drain current as output under continuous pressure loading at 200 kPa.

Revised Manuscript (Page 3):

The realization of artificial tactile sensations in electronic devices along with audiovisual technology can provide a more realistic user experience¹⁻⁵.

Revised Manuscript (Page 7):

We employed a pressure-sensitive field-effect transistor (FET) as a pressure sensor due to its superior capabilities in detecting and calibrating inter-individual pressure variations. Notably, the FET's potential for extension to high-dimensional devices offers significant advantages for integration with screen devices. A comparative analysis of our sensor's performance with previously reported motion sensors is provided in Supplementary Table 2.

Revised Manuscript (Page 8):

Furthermore, to examine the sensitivity of pressure-sensitive FET, real-time detection of the change in drain current was conducted during applying the external compressive pressure from 5 kPa to ~~300250~~ kPa (Fig. 1f and Supplementary Fig. 7). The sensitivity was calculated as $7.46 \times 10^{-3} \text{ kPa}^{-1}$ ~~0.0067~~ kPa^{-1} within the pressure range from 1 to 75 kPa, and 1.22×10^{-3} ~~0.0017~~ kPa^{-1} within the pressure range from 75 to ~~300250~~ kPa (Fig. 1g).

Revised Manuscript (Page 8):

Furthermore, the relative changes in the drain current of pressure-sensitive FET remained consistent, with the decrease being less than 1%, over 1000 cycles of compressive pressure loadings at 200 kPa (Supplementary Fig. 8). During this pressure loading, the response time and recovery time were examined to be 21 ms and 23 ms, respectively (Supplementary Fig. 9).

Revised Manuscript (Page 55):

5. Shi, Y. *et al.* Eye tracking and eye expression decoding based on transparent, flexible and ultra-persistent electrostatic interface. *Nat. Commun.* **14**, 3315 (2023).

Revised Supplementary Materials (Page 40):

Supplementary Fig. 7. Real-time detection of relative changes in drain current at (A) low-pressure ranges (<75 kPa), (B) and high-pressure ranges (>75 kPa, <300 kPa). Each step corresponds to 5, 15, 30, 45, 60, 75, 90, 105, 120, 135, 150, 165, 180, 195, 210, 225, 240, 255, 270, 285, and 300 kPa.

Revised Supplementary Materials (Page 41):

Supplementary Fig. 8. Reliability test during repetitive loading-unloading of pressure at 200 kPa with 1000 cycles.

Revised Supplementary Materials (Page 42):

Supplementary Fig. 9. Response time and recovery time during pressure loading at 200 kPa.

Revised Supplementary Materials (Page 82):

Sensor's Type	Detection target	Detection Range	Response time	Array size	Ref.
Capacitive	Finger pressure	~ 100 kPa	190 ms	1	37
	Various human motions	0 ~ 600 kPa	280 ms	1	38
	Eye movement	5 ~ 40°	-	4	39
Piezoelectric	Wrist pulse	0.125 ~ 22.5 kPa	18.6 ms	3	40
	Arterial pulse	1 ~ 60 kPa	60 ms	1	41
	Finger pressure	0 ~ 25 kPa	300 ms	9 (3 × 3)	42
Pressure-sensitive transistor	Wrist pulse	8 ~ 1500 Pa	< 10 ms	25 (5 × 5)	43
	Cellular motion	200 Pa ~ 5 MPa	25 ms	400 (20 × 20)	44
	Finger pressure (this work)	5 ~ 300 kPa	21 ms	100 (10 × 10)	-

Supplementary Table 2. Comparison in performances of three types of pressure sensors utilized for motion detection.

Revised Supplementary Materials (Page 89):

37. Lee, S. *et al.* Nanomesh pressure sensor for monitoring finger manipulation without sensory interference. *Science* **370**, 966–970 (2020).
38. He, X. *et al.* Microstructured capacitive sensor with broad detection range and long-term stability for human activity detection. *npj Flex. Electron.* **5**, 1–9 (2021).
39. Shi, Y. *et al.* Eye tracking and eye expression decoding based on transparent, flexible and ultra-persistent electrostatic interface. *Nat. Commun.* **14**, 3315 (2023).
40. Chu, Y. *et al.* Human Pulse Diagnosis for Medical Assessments Using a Wearable Piezoelectret Sensing System. *Adv. Funct. Mater.* **28**, 1803413 (2018).

41. Park, D. Y. *et al.* Self-Powered Real-Time Arterial Pulse Monitoring Using Ultrathin Epidermal Piezoelectric Sensors. *Adv. Mater.* **29**, 1702308 (2017).
42. Yu, H. *et al.* Flexible temperature-pressure dual sensor based on 3D spiral thermoelectric Bi₂Te₃ films. *Nat. Commun.* **15**, 2521 (2024).
43. Huang, Y.-C. *et al.* Sensitive pressure sensors based on conductive microstructured air-gap gates and two-dimensional semiconductor transistors. *Nat. Electron.* **3**, 59–69 (2020).
44. Jang, J. *et al.* Mechanoluminescent, Air-Dielectric MoS₂ Transistors as Active-Matrix Pressure Sensors for Wide Detection Ranges from Footsteps to Cellular Motions. *Nano Lett.* **20**, 66–74 (2020).

Comment 1: Besides, there are still some unsolved issues with this article, and I believe it needs to undergo revisions before it can be accepted.

It is usually difficult to balance touch and pain sensation in electrotactile systems. Can data or other quantifiable strategies be proposed to address this issue?

Response to Comment 1:

We appreciate the reviewer's thoughtful comments. As noted, it is difficult to balance touch and pain sensation in electrotactile systems. To enhance the dataset, we conducted supplementary experiments on the perception of electrotactile sensation, by applying current density with a finer step size between the levels of "Well perceptible" and "Pain". The results of this experiment revealed that the threshold current density marking the differentiation between the "Pain" and "Very well perceptible" levels is 0.0575 A mm⁻², with the onset of the "Pain" level being detected at 0.0638 A mm⁻² (Supplementary Note Fig. 9A).

In order to establish a clear distinction between pain and touch sensation and to develop a quantifiable strategy, Visual Analog Scale (VAS) test was conducted ranging from 0.05 A mm⁻² to 0.1 A mm⁻². The VAS test is a method for assessing pain in humans, enabling the representation of pain intensity through a numerical score on a pain scale. The VAS test classifies pain levels within the ranges of 4-6 as "Moderate pain", 7-8 as "Severe pain", and 9-10 as "Worst pain possible," a system endorsed by the World Health Organization for indicating pain severity in its guidelines for analgesic use [*Pediatr Nurs*, **14** (1), 9-17 (1988)]. In this test, "Moderate pain" indicates a level of irritation but not significant discomfort (Supplementary Note Fig. 9B).

A VAS test was administered alongside an electrotactile sensation intensity assessment to five subjects (aged 25-30; three males and two females) with the finger pressure calibrated to medium (100 kPa). Following the establishment of a VAS rate of 5 for the electrotactile sensation corresponding to "Very well perceptible" level, we recorded the responses when the current density was increased from 0.05 A mm⁻² to 0.1 A mm⁻². As a result, at a peak current density of 0.075 A mm⁻², which had been the highest level in the preceding experiment, the VAS test yielded an average response of 7 denoting "Severe pain," while at 0.1 A mm⁻², the response escalated to 10, indicating "Worst pain possible" (Supplementary Note Fig. 9C). While this experiment facilitated the representation of pain levels on a scale, the data obtained was inherently subjective, as it relies on the subject's response. Therefore, it is challenging to claim that it is accurate quantifiable data.

Therefore, we re-measured Somatosensory Evoked Potential (SEP), the baseline for electrotactile sensation, in three subjects (aged 25-30; two males and one female). Initially, SEP were obtained at 10 mA, a level at which no pain sensation is reported. Subsequently, data were gathered and analyzed for SEP ranging from 12 mA to 20 mA, representing the onset of the "Very well perceptible" level. The indicators utilized to correlate pain and touch sensation with SEP were the emergence, latency, and amplitude of P1-2 and N1-2 corresponding to the initial peak [*Exp. Brain Res.*, **84**, 607-619 (1991); *Neuroimage*, **146**, 575-588 (2017); *Neurosci. Lett.*, **647**, 110-116 (2017)]. The results showed that P1 was observed at 10 mA and 12 mA around 1-2 ms, while N2 was observed from 13 mA onwards. The latency of P1 at 10 mA and 12 mA remained relatively stable at 1.583 ± 0.057 ms and 1.567 ± 0.505 ms, respectively, although the amplitude exhibited a decrease from 7372 ± 42.14 μV to 1828 ± 333.16 μV. Moreover, within the range of 13 mA to 16 mA, the latency of N2 decreased from 2.717 ± 0.058 ms to 2.033 ± 0.029 ms, accompanied by an increase in amplitude from 2511 ± 270.022 μV to 3209 ± 259.356 μV. Conversely, for the 17 mA to 20 mA range, the latency of N1 notably decreased from 1.85 ± 0.13 ms to 1.583 ± 0.029 ms, and the amplitude tended to increase gradually from 933 ± 666.27 μV, which was lowered once, to 1957 ± 371.15 μV, 2048 ± 121.20 μV, and 2693 ± 108.01 μV (Supplementary Note Fig. 9D).

Following the integration of the quantitative analysis of SEP with the outcomes of the VAS test, the analysis revealed that the pain sensation could be categorized into three segments based on the presence or absence of N2: sensation below 13 mA, sensation below 16 mA with no discernible amplitude trend, and sensation up to 20 mA with a pattern of amplitude fluctuation (Supplementary Note Fig. 9E-F). By aligning the SEP data with the current density and electro tactile sensation data obtained from the prior VAS test, a clear differentiation between tactile sensation and pain was achievable. Ranging from 0.055 A mm^{-2} to 0.065 A mm^{-2} was identified as “Moderate pain”, 0.065 A mm^{-2} to 0.085 A mm^{-2} as “Severe pain”, and 0.085 A mm^{-2} to 0.1 A mm^{-2} as “Worst pain possible” (Supplementary Note Fig. 9G).

In conclusion, we successfully differentiated between pain and touch sensations based on current density levels. Through the analysis of VAS and SEP, it was determined that the pain intensity experienced at a current density of 0.075 A mm^{-2} , corresponding to the maximum current density in the original experiment, was classified as “Severe pain”, representing a relatively low level of actual pain. Moreover, given that touch perception within “Perceptible” to “Well perceptible” level is adequate for a realistic tactile simulation, a current density of 0.05 A mm^{-2} or lower was utilized in all applications to ensure subjects were exposed only to non-painful stimulation. All of these experiments are summarized in the manuscript and Supplementary Materials.

Revised Manuscript (Page 15):

Based on these results, all subsequent experiments were pressure-calibrated and performed at safe current densities, which were confirmed by additional experiments to distinguish between pain and touch sensation. In particular, distinguishing between pain and touch sensations is essential to safely implementing tactile sensations, so additional Visual Analog Scale and SEP experiments were conducted and compared again to determine the precise distinction between pain and sensation, as well as the stage of pain (Supplementary Note 8). ~~All subsequent experiments were pressure-calibrated based on these results.~~ This meticulous calibration ensured the reliability and accuracy of the experimental outcomes. In addition, the skin-electrode impedance changes with not only the pressure but also the humidity of skin, it is important to control the humidity for consistent results. Therefore, another factor that affects electro tactile sensation, skin humidity, was also controlled and adjusted to maintain consistent conditions throughout the tests under varying skin moisture conditions (Supplementary Note 9).

Revised Manuscript (Page 38):

Experiment 10: VAS test and SEP signal analysis for pain assessment

A VAS test was administered alongside an electro tactile sensation intensity assessment to five healthy adult subjects (age, 25–30 years; three males and two females) with the finger pressure calibrated to medium (100 kPa). Following the establishment of a VAS rate of 5 electro tactile sensation corresponding to “Very well perceptible” level, we recorded the VAS responses when the current density was increased from 0.05 A mm^{-2} to 0.1 A mm^{-2} . Also, we re-measured SEP, the baseline for electro tactile sensation, in three subjects (aged 25-30; two males and one female). Initially, SEP were obtained at 10 mA, a level at which no pain sensation is reported. Subsequently, data were gathered and analyzed for SEP ranging from 12 mA to 20 mA. The indicators utilized to correlate pain and touch sensation with SEP were the emergence, latency, and amplitude of P1-2 and N1-2 corresponding to the initial peak. This experiment was repeated once for each subject.

Revised Supplementary Materials (Page 20-24):

Supplementary Note 8. A quantifiable strategy for distinguishing between touch and pain sensation in an electro tactile system.

Supplementary Note Fig. 9. Experiments for distinguishing and analyzing the differentiation between pain and touch sensation. **(A)** Tactile intensity according to current density from 0.03 A mm⁻² to 0.075 A mm⁻² with a spacing of 0.0125 A mm⁻². **(B)** Tool used for pain assessment in the Visual Analog Scale (VAS) test. Scale bar = 1 cm **(C)** VAS test results from 0.05 A mm⁻² to 0.1 A mm⁻² **(D)** SEPs ranging from 0 ms to 5 ms with amplitudes of 10, 12, 13, 14, 15, 16, 17, 18, 19, and 20 mA (Average count for one signal = 300, Frequency = 4.7 Hz). **(E)**

Comparison of SEP amplitude of N2 and P1 with VAS results according to SEP current changes **(F)** Comparison of SEP latency of N2 and P1 with VAS results according to SEP current changes **(G)** Graph showing the differentiation of pain level based on changes in VAS results, SEP amplitude, and SEP latency when current density changes.

Additional experiments were conducted to effectively distinguish between touch and pain sensations and to control unpleasant electro tactile sensations. Initially, in order to ascertain the precise threshold for the sensation of "Pain" in electro tactile perception, we conducted a reassessment of the intensity of electro tactile sensation ranging from the point of being "Well perceptible" to that of "Pain" by reducing the current density interval. The stimulation parameters employed were a frequency of 10 Hz and a pulse width of 10 ms. As a result, the threshold current density marking the differentiation between the "Pain" and "Very well perceptible" levels is 0.0575 A mm^{-2} , with the onset of the "Pain" level being detected at 0.0638 A mm^{-2} (Supplementary Note Fig. 9A).

In order to establish a clear distinction between pain and touch sensation and to develop a quantifiable strategy, Visual Analog Scale (VAS) test was conducted. The VAS test is a method for assessing pain in humans, enabling the representation of pain intensity through a numerical score on a pain scale. The VAS test classifies pain levels within the ranges of 4-6 as "Moderate pain", 7-8 as "Severe pain", and 9-10 as "Worst pain possible," a system endorsed by the World Health Organization for indicating pain severity in its guidelines for analgesic use¹⁸. Based on the current density collected in the previous experiment, a VAS test was performed with stimuli ranging from 0.05 A mm^{-2} to 0.1 A mm^{-2} . In this test, "Moderate pain" indicates a level of irritation but not significant discomfort (Supplementary Note Fig. 9B).

A VAS test was administered alongside an electro tactile sensation intensity assessment to five subjects (age, 25-30 years; three males and two females) with the finger pressure calibrated to medium (100 kPa). Following the establishment of a VAS rate of 5 for the electro tactile sensation corresponding to "Very well perceptible" level, we recorded the responses when the current density was increased from 0.05 A mm^{-2} to 0.1 A mm^{-2} . As a result, at a peak current density of 0.075 A mm^{-2} , which had been the highest level in the preceding experiment, the VAS test yielded an average response of 7 denoting "Severe pain," while at 0.1 A mm^{-2} , the response escalated to 10, indicating "Worst pain possible" (Supplementary Note Fig. 9C). The experiment facilitated the representation of pain levels on a scale, but since the VAS test also relies on the subjective assessment of the subjects, we re-measured the SEPs that preceded it to establish a baseline of electro tactile sensation in order to collect accurate quantifiable data.

SEP data was first measured at 10 mA, where no pain was felt, and then SEP data was collected and compared from 12 mA to 20 mA, where the very well perceptible phase began with three subjects (age, 25-30 years; two males and one female). The experimental conditions and environment are the same as in experiment 2, except for the average count in Method. The average count used for a single collection was upscaled to 300 for tighter results. The indicators utilized to correlate pain and touch sensation with SEP were the emergence, latency, and amplitude of P1-2 and N1-2 corresponding to the initial peak¹⁹⁻²¹. The results showed that P1 was observed at 10 mA and 12 mA around 1-2 ms, while N2 was observed from 13 mA onwards. The latency of P1 at 10 mA and 12 mA remained relatively stable at $1.583 \pm 0.057 \text{ ms}$ and $1.567 \pm 0.505 \text{ ms}$, respectively, although the amplitude exhibited an decrease from $7372 \pm 42.14 \text{ } \mu\text{V}$ to $1828 \pm 333.16 \text{ } \mu\text{V}$. Moreover, within the range of 13 mA to 16 mA, the latency of N2 decreased from $2.717 \pm 0.058 \text{ ms}$ to $2.033 \pm 0.029 \text{ ms}$, accompanied by an increase in amplitude from $2511 \pm 270.022 \text{ } \mu\text{V}$ to $3209 \pm 259.356 \text{ } \mu\text{V}$. Conversely, for the 17 mA to 20 mA range, the latency of N1 notably decreased from $1.85 \pm 0.13 \text{ ms}$ to $1.583 \pm 0.029 \text{ ms}$, and the amplitude tended to increase gradually from $933 \pm 666.27 \text{ } \mu\text{V}$, which was lowered once, to $1957 \pm 371.15 \text{ } \mu\text{V}$, $2048 \pm 121.20 \text{ } \mu\text{V}$, and $2693 \pm 108.01 \text{ } \mu\text{V}$ (Supplementary Note Fig. 9D).

Following the integration of the quantitative analysis of SEP with the outcomes of the VAS test, the analysis revealed that the pain sensation could be categorized into three segments based on the presence or absence of N2: sensation below 13 mA, sensation below 16 mA with no discernible amplitude trend, and sensation up to 20 mA with a pattern of amplitude fluctuation (Supplementary Note Fig. 9E-F). By aligning the SEP data with the current density and electro tactile sensation data obtained from the prior VAS test, a clear differentiation between tactile sensation and pain was achievable. Ranging from 0.055 A mm^{-2} to 0.065 A mm^{-2} was identified as "Moderate pain", 0.065 A mm^{-2} to 0.085 A mm^{-2} as "Severe pain", and 0.085 A mm^{-2} to 0.1 A mm^{-2} as "Worst pain possible" (Supplementary Note Fig. 9G).

In conclusion, the differentiation between pain and touch sensation based on current density was successfully achieved. Based on the VAS and SEP analysis, it was determined that the pain sensation at a current

density of 0.075 A mm^{-2} , which corresponds to the existing "Pain" level, was at the "Severe pain" level, which is the low level of actual pain. Also, given that touch perception within "Perceptible" to "Well perceptible" level is adequate for a realistic tactile simulation, a current density of 0.05 A mm^{-2} or lower was utilized in all applications to ensure subjects were exposed only to non-painful stimulation.

Revised Supplementary Materials (Page 87):

19. Özgül, Ö. S. *et al.* High test-retest-reliability of pain-related evoked potentials (PREP) in healthy subjects. *Neurosci. Lett.* **647**, 110–116 (2017).
20. Bruyns-Haylett, M. *et al.* The neurogenesis of P1 and N1: A concurrent EEG/LFP study. *NeuroImage* **146**, 575–588 (2017).
21. Cauller, L. J. & Kulics, A. T. The neural basis of the behaviorally relevant N1 component of the somatosensory-evoked potential in SI cortex of awake monkeys: evidence that backward cortical projections signal conscious touch sensation. *Exp. Brain Res.* **84**, (1991).

Comment 2: Concerning the safety of the device during long time operation, is the electrical stimulation it employs completely harmless to human skin and tactile nerve?

Voltage leakage during operation may result in personal injury and what measurements have been taken to prevent such occurrences?

Response to Comment 2:

We thank the reviewer for providing a valuable comment. As the reviewer suggested, when electrical stimulation is applied for a long period of time, it is crucial to investigate the potential effects on human nerves and determine if cell death occurs. Therefore, we conducted experiments to test the overall safety of the device during long-term operation. Safety tests were conducted on human skin, animal skin, and human nerve cells. The experiments were divided into passive state (control) when no stimulation was applied and active state when stimulation was applied.

Initially, we compared the effects of applying electrical stimulation for one hour with no stimulation on the skin of the arm using the transparent pressure-calibratable interference electro-tactile actuator (TPIEA) device. The stimulation parameters included a frequency of 10 Hz, pulse width of 10 ms, and intensity of 0.015 A mm^{-2} . Results indicated the absence of skin irritation or redness upon removal of the device in both the control and stimulation states, as illustrated in Supplementary Note Fig. 2.

Subsequently, following the application of stimulation on the back of mouse using the TPIEA device, epidermal samples were collected, fixed in paraffin, and subjected to hematoxylin and eosin (H&E) staining to assess changes in dermis-epidermis thickness and the presence of burn marks. Stimulation was administered for 6 minutes, comprising 5 minutes of electrical stimulation and 1 minute of rest, with experiments conducted under various durations (no stimulation, 1 set - 6 minutes, 5 sets - 30 minutes, and 10 sets - 60 minutes). The stimulation parameters remained consistent at 10 Hz frequency, 10 ms pulse width, and 0.015 A mm^{-2} . As shown in the Supplementary Note Fig.3, H&E staining revealed the architecture of epidermis, dermis, hair follicles and muscles of each group did not under the extent of 0.015 A mm^{-2} stimulation (compare double arrow of Supplementary Note Fig. 3F and 3I).

Thirdly, we tested cell viability at different time points when electrical stimulation was applied using the Calcein AM assay. We cultured SH-SY5Y cells, which are a human neuroblastoma cell line, on the device and applied electrical stimulation with a current intensity of 0.015 A mm^{-2} , a frequency of 10 Hz, and a pulse width of 10 ms for 0 minute (control group), 5 minutes, 30 minutes, 60 minutes, and 180 minutes. As a result of Calcein AM, fluorescence images captured with an optical microscope exhibit consistent trends for each group (Supplementary Note Fig. 4.). After applying electrical stimulation for 180 minutes, only a few cells died, and overall cell viability was maintained.

Lastly, for a quantitative analysis, fluorescence intensity was calculated from the Calcein AM results. To assess cell death, 70% ethanol was used as a positive control group. Supplementary Note Fig. 5. shows the results for each group, with the viability of the ethanol-treated group (positive control), 0 minute (control), 30 minutes, 60 minutes, and 180 minutes of electrical stimulation as $33.96 \pm 2.98\%$, 100%, $95.09 \pm 5.69\%$, $92.32 \pm 1.65\%$, and $94.21 \pm 5.04\%$, and $93.74 \pm 1.43\%$, respectively. The experiments were performed with a sample size of 3 for each assay. The fluorescence intensity remained above 93% after electrical stimulation, regardless of the duration. The safety of TPIEA's electrical

stimulation was validated through these three experiments, indicating the absence of adverse impacts on human skin or tactile nerves following prolonged exposure to electrical stimulation.

Also, we thank reviewer for providing valuable comment regarding the potential safety concerns associated with voltage leakage when applying electrical stimulation. To address these concerns and ensure the safety of our experimental setup, we implemented a method to effectively manage and control voltage leakage. As illustrated in Figure R1, we could connect the pulse generator to the device through a SourceMeter to limit the applied voltage.

Figure R1. The pulse generator is connected to the device via a SourceMeter to manage and control voltage leakage.

The principle behind this setup involves the SourceMeter acting as an intermediary between the pulse generator and the device, enabling us to apply a compliance voltage limit. The SourceMeter is configured with a compliance limit, which is a preset threshold that restricts the maximum voltage and current that can pass through the circuit. This prevents any excessive voltage from reaching the device, thereby reducing the risk of voltage leakage and ensuring the safety of both the device and the user. The SourceMeter continuously measures the voltage and current in real time. If the supplied voltage or current exceed the compliance limit, the SourceMeter automatically adjusts or shuts down the supply to prevent any potential hazards. By placing the SourceMeter between the pulse generator and the device, we can regulate the pulse characteristics with high precision. This setup ensures that the electrical stimulation is within the safe operational parameters defined for the experiment. This approach effectively addresses the reviewer's concern by providing a robust safety mechanism against voltage leakage.

In addition, additional computations were performed to demonstrate that even in the event of voltage leakage, the current being utilized remains at a safe level. Voltage leakage can potentially result in personal harm if excessive current flows through the human body due to such leakage. As indicated in Table R1, problematic levels of leakage current commence at 10 mA, posing a risk of medical injury.

Current (mA)	Response
0.2-2	“Electrical” sensation
1-2	Painful shock
3-5	Let-go threshold for children

6-10	Minimum let-go threshold for adults
22	99% of adults cannot let go
10-20	Tetany (contact area)
20-50	Tetany (respiratory muscles)
50-100	Ventricular fibrillation

Table R1. Physical Effects at selected leakage currents [Percept. Psychophys., **42** (3), 257-268 (1987); Electrical Shock Safety Criteria, *Physiological Effects of Electric Currents on Living Organisms, more particularly Humans*, 7-24, 1985, Pergamon]

The formula for leakage current, $I = 377VC$, was employed to determine the current leakage that would arise in the event of device malfunction. As explained in Comment 3, the total maximum operating voltage calculated from the current density is assumed to be 150 V. With a measured capacitance of 31.17 nF for the device, the maximum leakage current was calculated to be 1.7 mA (Supplementary Note Fig. 6). Based on this analysis, it has been established that the experiment was conducted within a safe current threshold, as the potential leakage current within the experiment remains below 2 mA. These results have been incorporated into the supplementary materials and the manuscript.

Revised Manuscript (Page 9):

In addition to these electrical performances, the biocompatibility and safety of the TPIEA material were rigorously assessed. Various calculations, including adherence to international general standards, skin compatibility, and cell viability tests, confirmed its suitability for potential biomedical applications (Supplementary Notes 3 and 4).

Revised Manuscript (Page 29-30):

Impedance and charge storage capacity spectroscopy

The pristine ITO and ITO-Pt electrodes were assessed for impedance and charge storage capacity spectroscopy in a DPBS solution (Sigma-Aldrich). Impedance measurements were carried out using a multichannel potentiostat (PMC-1000, AMETEK) within a frequency range of 0.01 to 100 kHz. Charge storage capacity was determined at a rate of 100 mV s⁻¹ and the area was computed through graph analysis. Capacitance was determined at a frequency of 1000 Hz with 25 data points. To assess electrical fatigue, electrical stimulation was administered at a current density of 0.01 A mm⁻², a frequency of 10 Hz, and a pulse width of 10 ms, followed by the measurement of electrode impedance within a 1 kHz bandwidth. The impedance was assessed three times, with stimulation durations of 5, 10, 20, 30, 40, 50, 60, and 70 minutes.

Revised Manuscript (Page 30-31):

Cell viability test

We cultured SH-SY5Y cell line, which is a human neuroblastoma cell line, on the device and applied electrical stimulation with a current intensity of 0.015 A mm⁻², a frequency of 10 Hz, and a pulse width of 10 ms for 0 minute (control group), 5 minutes, 30 minutes, 60 minutes, and 180 minutes. As a result of Calcein AM, fluorescence images confirmed with an optical microscope exhibit consistent trends for each group.

Evaluation of skin safety in mice during long-term electrical stimulation

The experiment with mouse was performed in compliance with all the ethical regulations under a protocol that was approved by the Institutional Animal Care and Use Committee of Yonsei University (application no. IACUC-A-202405-1851-01). C57BL/6 male mice (6-8 weeks old) were purchased from Orient Bio (Seoul, Korea). The animals were maintained on a 12/12-hour light/dark cycle in a temperature-controlled room (23 ± 2°C). The skin of each group of mice was removed the hair with an electric razor and hair removal gel. The mouse skin tissue was cut (0.8 cm × 0.8 cm) and fixed in 4% paraformaldehyde for 24 h at 4 °C after stimulation. Stimulation was administered for 6 minutes, comprising 5 minutes of electrical stimulation and 1 minute of rest, with experiments conducted under varying durations (no stimulation, 1 set - 6 minutes, 5 sets - 30 minutes, and 10 sets - 60 minutes). The stimulation parameters remained consistent at 10 Hz frequency, 10 ms pulse width, and 0.015 A mm⁻². They were embedded in paraffin and sectioned at thickness of 5 µm. Following deparaffinization and rehydration, sections were stained with an H&E stain kit according to the protocol provided by Vector (Brussels, Belgium). The image of the section was obtained under a microscope (Olympus BX53M).

Revised Manuscript (Page 37):

Experiment 9: Observation of the skin state when using TPIEA

The stimulation parameters included a frequency of 10 Hz, pulse width of 10 ms, and intensity of 0.015 A mm^{-2} on skin. After 1 hour of stimulation, one subject (age, 27 years; one female) was instructed to compare the skin to non-stimulated skin to ensure that no skin irritation or redness has occurred.

Revised Supplementary Materials (Page 5-8):

Supplementary Note 3. The safety of TPIEA's electrical stimulation on skin and tactile nerve

Supplementary Note Fig. 2. Safety of TPIEA on human skin. Photographs of skin condition before stimulation and skin condition without (passive state) and with electrical stimulation (active state) after 1 hour. Scale bar, 2 cm.

Initially, we compared the effects of applying electrical stimulation for one hour with no stimulation on the skin of the arm using the TPIEA device. The stimulation parameters included a frequency of 10 Hz, pulse width of 10 ms, and intensity of 0.015 A mm^{-2} . Results indicated the absence of skin irritation or redness upon removal of the device in both the control and stimulation states, as illustrated in Supplementary Note Fig. 2.

Supplementary Note Fig. 3. H&E staining showing the histological structure of the mouse dorsal skin. The structure of the epidermis in the (A) dorsal skin of mice. Scale bar = 1 cm. (B, F) control; wo (C, G) 1 times (D, H) 5 times and (E, I) 10 times after 0.015 A mm^{-2} stimulation.

Subsequently, following the application of stimulation on the back of mouse using the TPIEA device, epidermal samples were collected, fixed in paraffin, and subjected to hematoxylin and eosin (H&E) staining to assess changes in dermis-epidermis thickness and the presence of burn marks. Stimulation was administered for 6 minutes, comprising 5 minutes of electrical stimulation and 1 minute of rest, with experiments conducted under varying durations (no stimulation, 1 set - 6 minutes, 5 sets - 30 minutes, and 10 sets - 60 minutes). The stimulation parameters remained consistent at 10 Hz frequency, 10 ms pulse width, and 0.015 A mm^{-2} . The stimulation parameters remained consistent at 10 Hz frequency, 10 ms pulse width, and 0.015 A mm^{-2} . As shown in the Supplementary Note Fig.3, H&E staining revealed the architecture of epidermis, dermis, hair follicles and muscles of each group did not under the extent of 0.015 A mm^{-2} stimulation (compare double arrow of Supplementary Note Fig. 3F and 3I).

Supplementary Note Fig. 4. Calcein AM assays of SH-SY5Y cells cultured in media containing the tactile devices for 0 minutes, 5 minutes, 30 minutes, 60 minutes, and 180 minutes of electrical stimulation (ES). All scale bars = 100 μ m.

Third, we tested cell viability at different time points when electrical stimulation was applied using the Calcein AM assay. We cultured SH-SY5Y cells, which are a human neuroblastoma cell line, on the device and applied electrical stimulation with a current intensity of 0.015 A mm^{-2} , a frequency of 10 Hz, and a pulse width of 10 ms for 0 minute (control group), 5 minutes, 30 minutes, 60 minutes, and 180 minutes. As a result of Calcein AM, fluorescence images confirmed with an optical microscope exhibit consistent trends for each group (Supplementary Note Fig. 4). After applying electrical stimulation for 180 minutes, only a few cells died, and overall cell viability was maintained.

Supplementary Note Fig. 5. Fluorescence intensity of Calcein AM based on the control group (0 minute) ($n=3$). To compare cell death levels, a positive control group treated with 70% ethanol was included. Significant differences were analyzed with the unpaired student's t-test and marked as ns ($p > 0.05$), ** ($p < 0.01$) and *** ($p < 0.001$). The exact p values are, from left to right, 6.218×10^{-6} , 2.75×10^{-3} , and 3.425×10^{-3} .

In addition, for quantitative analysis, fluorescence intensity was calculated from the calcein-acetoxymethyl results. To assess cell death, 70% ethanol was used as a positive control group. Supplementary Note Fig. 5 shows the results for each group, with the viability of the ethanol-treated group (positive control), 0 minute (control), 30 minutes, 60 minutes, and 180 minutes of electrical stimulation as $33.96 \pm 2.98\%$, 100%, $95.09 \pm 5.69\%$, $92.32 \pm 1.65\%$, and $94.21 \pm 5.04\%$, and $93.74 \pm 1.43\%$, respectively. The experiments were performed with a sample size of 3 for each assay. The fluorescence intensity remained above 93% after electrical stimulation, regardless of the duration.

The safety of TPIEA's electrical stimulation was validated through these three experiments, indicating the absence of adverse impacts on human skin or tactile nerves following prolonged exposure to electrical stimulation.

Revised Supplementary Materials (Page 11-12):

(3) Evaluation of the leakage current level.

Supplementary Note Fig. 6. Capacitance of TPIEA device.

Furthermore, additional computations were performed to demonstrate that even in the event of voltage leakage, the applied current remains at a safe level. Voltage leakage can potentially result in personal injury if excessive current flows through the human body due to such leakage. The problematic levels of leakage current commence at 10 mA, posing a risk of medical injury^{10,11}. The formula for leakage current, $I = 377VC$, was employed to determine the current leakage that would arise in the event of device malfunction. As explained in Comment 3, the total maximum operating voltage calculated from the current density is assumed to be 150 V. With a measured capacitance of 31.17 nF for the device, the maximum leakage current was calculated to be 0.00176436 A, corresponding to 1.7 mA (Supplementary Note Fig. 6). Based on this analysis, it has been established that the experiment was conducted within a safe current threshold, as the potential leakage current within the experiment remains below 2 mA.

Revised Supplementary Materials (Page 86):

10. Cabanes, J. *Electrical Shock Safety Criteria Ch.1* (Pergamon, Oxford, 1985).

11. Rollman, G. B. & Harris, G. The detectability, discriminability, and perceived magnitude of painful electrical shock. *Percept. Psychophys.* **42**, 257–268 (1987).

Comment 3: Repeated stimulation can lead to a decrease in a person's perceptual ability and the long-time electrical stimulation at the same location may cause nerve paralysis. What are the proposed solutions to mitigate or prevent this reduction in tactile perception? Has the fatigue test been conducted? The author needs to confirm whether the magnitude of the electrical stimulation current meets the general standards (for example, the GB-9706 standards for medical devices).

Response to Comment 3:

We thank reviewer for raising an insightful comment. In response to the reviewer's comment, we initially describe the absence of nerve paralysis resulting from prolonged electrical stimulation at a specific site, as well as strategies to mitigate the desensitization of tactile perception due to repetitive experimentation. Subsequently, a series of supplementary fatigue tests encompassing electrical fatigue, mechanical fatigue, and nerve fatigue evaluations were conducted to prove that extended electrical stimulation does not induce adverse effects on both subjects and device property. Lastly, the electrical stimulation current used in TPIEA was quantified and validated to comply with diverse international standards.

(1) Assessment of the safety of TPIEA

To prevent the potential reduction in sensitivity to tactile stimuli resulting from repeated testing, subjects were given

a 5-minute intermission following each 10-minute cognitive assessment. This interlude was incorporated into all cognitive evaluations involving the subjects, as detailed in the revised manuscript. Furthermore, an assessment of the current density utilized in the experiments was conducted to confirm the absence of nerve paralysis following prolonged application. In accordance with the recommendations outlined in the design principles for creating a safe system for applying current to the human body through a device [*ACM Trans. on Comput.-Hum. Interact.*, **25** (3), 1-36 (2018)], electro-tactile stimulation is identified as a haptic technique that falls within the spectrum of perceptual responses to electrical stimulation methods. By calculating the current intensity based on current density, it was determined that a current of up to 15 mA and 10 ms time duration could be administered in our study.

In accordance with the data presented in the AC current table (15-100 Hz) as outlined in the international standard [IEC 2016. IEC 60479-1: Effects of Current on Human Beings and Livestock. IEC.] (Figure R2), the current density observed in our study aligns with the AC-2 Perceptible stage.

Figure R2. Effects of AC current (15–100Hz) on the human body when it passes from the hand to the feet. (The figure is referenced from IEC TS 60479-1 ed.4.13 [IEC 2016. IEC 60479-1: Effects of Current on Human Beings and Livestock. IEC].)

This stage denotes a current level that does not induce muscular contraction or irreversible consequences. Furthermore, as indicated in response to the comment 2 above, the current intensity utilized is below the threshold that triggers nerve paralysis or muscular contraction, even in the presence of leakage current. Therefore, we have confirmed the safety of leakage current by stimulating with a current amount that is far below the current threshold that causes nerve paralysis.

(2) Electrical fatigue test of TPIEA

We also carried out supplementary fatigue assessments encompassing electrical, mechanical, and nerve fatigue evaluations. Firstly, the electrical fatigue examination validated that the operational attributes, impedance, and pressure sensitivity of the device remained consistent during prolonged operation. Electrical stimulation was administered at a current density of 0.01 A mm^{-2} , a frequency of 10 Hz, and a pulse width of 10 ms, followed by the measurement of electrode impedance within a 1 kHz bandwidth. The impedance was assessed three times, with stimulation durations of 5, 10, 20, 30, 40, 50, 60, and 70 minutes. The findings indicated an average impedance of $1.3203 \pm 0.125 \text{ } \Omega \text{ cm}^2$, which did not significantly deviate from the unstimulated state impedance of $1.2086 \text{ } \Omega \text{ cm}^2$, thereby indicating no discernible functional disparity as an actuator even under extended operational periods. Furthermore, the impedance was approximately four times lower compared to that of pristine ITO electrodes (Supplementary Fig. 6). Furthermore, to establish the durability of the device, it was demonstrated that the pressure sensing performance of the pressure-sensitive FET remained consistent after undergoing 1000 cycles of cyclic pressure loading up to 200 kPa. The relative changes in drain current were found to be stable throughout these repeated pressure loading cycles, with the output current decrease of the pressure-sensitive FETs measuring less than 1% ($\sim 0.4\%$) after 1000 cycles (Supplementary Fig.8).

(3) Mechanical fatigue test of TPIEA

Secondly, the mechanical fatigue tests were conducted to TPIEA device. The tensile strength of the device was found

to be 0.24 MPa, exhibiting a yield point at a low strain of 1.4815%. Following yielding, the device displayed a consistent stress level attributed to the presence of PDMS, an elastomeric substance. The flexural strength was measured at 61.7 MPa, with fracture occurring once the bend angle reached 1.213° after the loading pin had been displaced by 0.8192 mm. Additionally, an impact resistance test was conducted to meet the criteria of the Izod impact test with an IH1 pendulum as the impact source. The application of 3.7 kJ m^{-2} with the pendulum led to the complete breakage of the device. These mechanical fatigue tests encompassing tests for tensile strength, angle of bend, and impact resistance were conducted, with further elaboration provided in comment 4 (Supplementary Note Fig. 1).

(4) Nerve fatigue test of TPIEA

Lastly, to demonstrate the absence of nerve fatigue in individuals exposed to electrotactile sensation for a long period of time, prolonged electrotactile exposure experiment was conducted involving three subjects (aged 25-30, comprising 2 males and 1 female). The experiment involved repeating the electrotactile intensity test 30 times at medium finger pressure and measured the threshold current density (A) from the "Perceptible" level to the "Well perceptible" level, as well as the threshold current density (B) from the "Well perceptible" level to the "Very well perceptible" level. The findings revealed that, on average, the threshold current density (A) varies within the range of 0.006 A mm^{-2} and the threshold current density (B) within 0.0058 A mm^{-2} after 30 repetitions. It suggests that while the current density threshold fluctuated slightly with repeated trials, this fluctuation was lower than the threshold change induced by pressure (which could be as high as 0.035 A mm^{-2}) (Supplementary Fig. 24)

(5) General standards for current on human

Finally, to verify whether the magnitude of the electrical stimulation current meets the general standards, we referred IEC 60601-2-10 established by International Electrotechnical Commission, which is equivalent of GB-9706 standards for medical devices. IEC 60601-2-10 discusses particular requirements for the basic safety of nerve stimulation and section 104 of IEC 60601-2-10 clearly specifies the requirements of output energy for various wave types. Section 104 states: For pulse outputs, 1) the maximum energy per pulse shall not exceed 300 mJ when applied to a load resistance of $500 \ \Omega$ and 2) the maximum output voltage shall not exceed a peak value of 500 V, when measured under open circuit conditions. We verified whether our electrical stimulation meets these requirements. Firstly, we calculated maximum energy per pulse when load resistance of $500 \ \Omega$ is applied. Combining electrical energy expressed as: $E = P \cdot t$ and electrical power expressed as: $P = I^2 R$ yields $E = I^2 R \cdot t$. Using this formula, maximum energy per pulse was calculated. In our work, the maximum current density applied to skin is $0.075 \text{ A} \cdot \text{mm}^{-2}$. Therefore, multiplying the area of the electrode to maximum current density yields maximum current applied. As electrode of the device is in a circular shape and radius is $250 \ \mu\text{m}$, area is calculated as $A = \pi \cdot (250 \ \mu\text{m})^2 \approx 0.2 \text{ mm}^2$. Multiplying area of the electrode to maximum current density yields 0.015 A (15 mA). Substituting maximum current value to load resistance of $500 \ \Omega$ yields 0.1125 W , and multiplying 0.1125 W to pulse time yields 0.001125 J (1.125 mJ) and 0.05625 J (56.25 mJ) per pulse, which are maximum energy per pulse used in our work. Both pulses satisfy the general standard (300 mJ per pulse) established by IEC 60601-2-10, suggesting electrical stimulation does meet the general standard.

Secondly, whether electrical stimulation applied in our work meets the second criterion, the maximum output voltage shall not exceed a peak value of 500 V when measured under open circuit condition, was verified by calculating the maximum amount of output voltage flow through the during device operation. In our work, the maximum current density applied to skin is $0.075 \text{ A} \cdot \text{mm}^{-2}$. The maximum current applied can be calculated by multiplying the area of the electrode, which is calculated as $A = \pi \cdot (250 \ \mu\text{m})^2 \approx 0.2 \text{ mm}^2$. Using Ohm's law, $V = IR$, we can calculate the maximum voltage applied while the device is operating. Since skin is in contact with the device during electrical stimulation, resistance is expressed as series circuit ($R_{Device} + R_{skin}$). As shown in Figure 1d, impedance of the electrode is $1.11 \ \Omega \cdot \text{cm}^2$ and dividing the area of the electrode (0.002 cm^2) yields $555 \ \Omega$. In our work, electrical stimulation was performed for dry skin condition only, and resistance of the dry skin is expressed as $10 \text{ k}\Omega$. Thus, the total resistance is calculated as $10555 \ \Omega$. According to the Ohm's law, the maximum output voltage is $0.015 \text{ A} \cdot 10555 \ \Omega = 158.325 \text{ V}$. This calculation proves that electrical stimulation conducted in our work is safe even at the maximum output, which is well below 500 V.

In conclusion, we verified that electrical stimulation performed in our work satisfy safety requirements stated in IEC 60601-2-10, which is the international standard for medical electrical equipment established by International Electrotechnical Commission. The maximum energy per pulse of electrical stimulation utilized in our work was calculated as 1.125 mJ and 56.25 mJ, which is well below 300 mJ per pulse, meeting the standard value of IEC 60601-2-10. The maximum output voltage under open circuit was calculated as 158.325 V, which is considerably lower than

500 V, also meeting the standard value of IEC 60601-2-10. This confirms our work was conducted under conditions that meet the international standards. These findings have been incorporated into the supplementary materials and the manuscript.

Revised Manuscript (Page 7):

Mechanical properties such as tensile strength, flexural strength, and impact resistance of TPIEA were also examined to verify the stability of the device under repetitive use. Since the majority of the device's volume or area is occupied by the substrate, these mechanical properties are strongly influenced by the materials of the substrate (Supplementary Note 2).

Revised Manuscript (Page 7-8):

At 1 kHz frequency, for example, the ITO-Pt electrode exhibited an impedance of $1.11 \Omega \text{ cm}^2$ (red dot) while the pristine ITO electrode presented $5.95 \Omega \text{ cm}^2$ (black dot), about 5 times lower ~~higher~~-value (Fig. 1d and sSupplementary Fig. 5). Even after applying a 70-minute electrical stimulation, the ITO-Pt electrode shows an impedance value approximately four times lower than that of the pristine ITO electrode (Supplementary Fig. 6).

Revised Manuscript (Page 8):

Furthermore, the relative changes in the drain current of pressure-sensitive FET remained consistent, with the decrease being less than 1%, over 1000 cycles of compressive pressure loadings at 200 kPa (Supplementary Fig. 8). During this pressure loading, the response time and recovery time were examined to be 21 ms and 23 ms, respectively (Supplementary Fig. 9).

Revised Manuscript (Page 8-9):

In addition to these electrical performances, the biocompatibility and safety of the TPIEA material were rigorously assessed. Various calculations, including adherence to international general standards, skin compatibility, and cell viability tests, confirmed its suitability for potential biomedical applications (Supplementary Notes 3 and 4).

Revised Manuscript (Page 16):

In addition, the capability of repetitive use was verified through a nerve fatigue test involving 30 trials per person. The results showed no significant change in the intensity of electrotactile sensation (Supplementary Fig. 24).

Revised Manuscript (Page 30):

Characterization of pressure-sensitive transistor

The electrical performances of pressure-sensitive FET, including transfer and output characteristics, were characterized using a probe station (Keithley 4200-SCS). The transfer characterization was examined at a drain voltage of 1 V. The output characterization was evaluated with the gate voltage ranging from 20 to 70 V in 10 V increments. To assess the pressure-sensing performance, the pressure was applied to the sensor using a high-precision motorized z-stage (Mark-10 ESM303).

Revised Manuscript (Page 32):

All of the subjects prepared for dull sensations by having a 5-minute break time after conducting a 10-minute cognitive experiment to prevent the reduction in tactile perception.

Revised Manuscript (Page 38):

Experiment 11: Threshold of electrotactile sensation during long-term stimulation

Three healthy adult subjects (age, 25–30 years; two males and one female) were arranged in the test to evaluate the perception threshold with long-term stimulation. The experiment involved repeating the electrotactile intensity test 30 times at medium finger pressure and measured the threshold current density (A) from the "Perceptible" level to the "Well perceptible" level, as well as the threshold current density (B) from the "Well perceptible" level to the "Very well perceptible" level. This experiment was repeated ten times for each subject.

Revised Supplementary Materials (Page 4):

Supplementary Note 2. Mechanical fatigue tests of the TPIEA device

Supplementary Note Fig. 1. Mechanical properties of TPIEA device. (A) Stress-Strain curve and (B) flexural Stress-Strain curve of TPIEA device.

The mechanical characteristics of TPIEA are assessed across three primary classifications, namely tensile strength, flexural strength, and impact resistance. First, tensile strength test using a Universal Testing Machine (Instron 3367) with a 30,000 N load cell at 5 mm min^{-1} and a 50 mm grip distance determined the tensile strength as 0.24 MPa, with a yield strain of 1.4815%, indicative of a brittle specimen. Post-yield, constant stress was observed due to the elastomeric PDMS. Second, flexural strength test using a Universal Testing Machine (Instron 34SC-5) with a 5,000 N load cell at 0.7 mm min^{-1} and a 25 mm support distance determined the flexural strength as 61.7 MPa. Fracture occurred at a bend angle of 1.213° with a loading pin displacement of 0.8192 mm, indicative of a brittle material. The flexural strength and bend angle were significantly influenced by the glass substrate, similar to the tensile strength results. Lastly, impact resistance test, conforming to Izod impact test criteria, was conducted using a Digital Impact Tester (QM700A, Qmesys) with an IH1 pendulum. The test, which measures the energy required to break a standardized specimen, showed that applying 3.7 kJ m^{-2} resulted in the complete breakage of the device.

Revised Supplementary Materials (Page 9-12):

Supplementary Note 4. Assessing compliance with the safety regulations to TPIEA

The safety of TPIEA has been demonstrated through three key factors: (1) assessment of the current level employed, (2) adherence to established general standard criteria, and (3) evaluation of the leakage current level.

(1) Assessment of the current level employed.

To demonstrate that nerve paralysis does not occur for amount and duration of current applied in this work, evaluation of the current density used in the experiment was conducted. The maximum current applied was calculated through multiplying maximum current density applied to skin with the area of the electrode, which is expressed as *Current density* \times *Area of the electrode (A)*, where: $A = \pi \cdot (250 \text{ } \mu\text{m})^2 \approx 0.2 \text{ mm}^2$. The maximum current applied was calculated as 15 mA. According to the design guidelines of device for developing safe electrical current delivery system to the human body⁵, electrotactile system presented in this work is a haptic technique that corresponds to perception among the methods of electrical application. Moreover, according to the international standard⁶, the current density level in this work, which is applied for 10 ms time duration, corresponds precisely to the AC-2 Perceptible phase, which is the level of current that does not cause muscular contraction or irreversible clinical effects.

(2) Adherence to established general standard criteria.

To verify the compliance of the electrical stimulation presented in this work with international standards, IEC 60601-2-10, which is equivalent to GB-9706 standards for medical devices, was referred⁷. IEC 60601-2-10 outlines safety requirements for nerve stimulation, specifying that for pulse outputs, the maximum energy per pulse should not exceed 300 mJ with a $500 \text{ } \Omega$ load resistance, and the maximum output voltage should not exceed 500 V in open circuit conditions. The maximum energy per pulse was calculated using the formula $E = I^2 R t$. With a maximum current density of 0.075 A mm^{-2} applied to a circular electrode of $250 \text{ } \mu\text{m}$ radius (0.2 mm^2 area), the maximum current was determined to be 0.015 A (15 mA). For a $500 \text{ } \Omega$ load resistance, the maximum power was 0.1125 W, resulting in pulse energies of 1.125 mJ and 56.25 mJ, both well within the 300 mJ limit, confirming

compliance with IEC 60601-2-10 standards.

To verify compliance with the IEC 60601-2-10 standard's second criterion, which limits the maximum output voltage to 500 V under open circuit conditions, the maximum voltage applied during device operation was calculated using the Ohm's law ($V = IR$). The maximum current density applied was 0.075 A mm^{-2} , and the electrode area was approximately 0.2 mm^2 , resulting in a maximum current of 0.015 A (15 mA). The total resistance expressed as series circuit ($R_{Device} + R_{skin}$), combining the device impedance (555Ω) and dry skin resistance ($10 \text{ k}\Omega$), was calculated as $10,555 \Omega$. Using the Ohm's law ($V = IR$), the maximum output voltage was calculated as $0.015 \text{ A} \cdot 10555 \Omega = 158.325 \text{ V}$. This value is well below the 500 V limit, confirming the safety of the electrical stimulation utilized in this work.

In conclusion, conducted verification process confirmed that the electrical stimulation used in this study complies with the IEC 60601-2-10 safety requirements for medical electrical equipment, proving the safety of the TPIEA. The maximum energy per pulse was 1.125 mJ and 56.25 mJ , well below the 300 mJ limit. Additionally, the maximum output voltage under open circuit conditions was 158.325 V , significantly lower than the 500 V threshold. These results demonstrate that electrical stimulation presented in this work meet international safety standards.

In addition to IEC 60479-1, other international standards for a low-frequency electrotherapy equipment, namely Japanese Industrial Standards (JIS), JIS T 2003:2005⁸ and JIS C 9335-2-209:2007⁹, are also satisfied, which are stated as:

- The current must be lower than 20 mA (with $1 \text{ k}\Omega$ resistance).
- The frequency must be lower than $1,200 \text{ Hz}$.
- The voltage must be lower than 200 V (with $1 \text{ k}\Omega$ resistance)
- The pulse energy must be lower than 120 mJ (with $1 \text{ k}\Omega$ resistance).

Regarding the safety and reliability of the device, all cognitive experiments were performed under the supervision of a neurosurgeon, which is reflected in the manuscript.

(3) Evaluation of the leakage current level.

Supplementary Note Fig. 6. Capacitance of TPIEA device.

Furthermore, additional computations were performed to demonstrate that even in the event of voltage leakage, the applied current remains at a safe level. Voltage leakage can potentially result in personal injury if excessive current flows through the human body due to such leakage. The problematic levels of leakage current commence at 10 mA , posing a risk of medical injury^{10,11}. The formula for leakage current, $I = 377VC$, was employed to determine the current leakage that would arise in the event of device malfunction. As explained in Comment 3, the total maximum operating voltage calculated from the current density is assumed to be 150 V . With a measured capacitance of 31.17 nF for the device, the maximum leakage current was calculated to be 0.00176436 A , corresponding to 1.7 mA (Supplementary Note Fig. 6). Based on this analysis, it has been established that the experiment was conducted within a safe current threshold, as the potential leakage current within the experiment remains below 2 mA .

Revised Supplementary Materials (Page 39):

Supplementary Fig. 6. Change in impedance of the electrode as a function of stimulation time.

Revised Supplementary Materials (Page 41):

Supplementary Fig. 8. Reliability test during repetitive loading-unloading of pressure at 200 kPa with 1000 cycles.

Revised Supplementary Materials (Page 57):

Supplementary Fig. 24. The threshold current density from “Perceptible” level to “Well perceptible” level (A) and the threshold current density from “Well perceptible” level to “Very well perceptible” level (B) as a result of 30 repeated collections with three subjects.

Revised Supplementary Materials (Page 86):

10. Cabanes, J. *Electrical Shock Safety Criteria Ch.1* (Pergamon, Oxford, 1985).
11. Rollman, G. B. & Harris, G. The detectability, discriminability, and perceived magnitude of painful electrical shock. *Percept. Psychophys.* **42**, 257–268 (1987).

Comment 4: I found that the whole device is made of rigid materials. What level of tensile strength, angle of bend, and impact resistance can this device tolerate? Is it possible to construct the device to be completely flexible? (Adv. Funct. Mater. 2018, 28, 1802989).?

Response to Comment 4:

We thank reviewer’s valuable comment. To respond to reviewer’s comment, we conducted evaluation of mechanical properties of TPIEA, including tensile strength, flexural strength, and impact resistance test of our device under ambient condition with 23 ± 2 °C and 45 ± 5 % relative humidity.

Firstly, tensile strength test was conducted using Universal Testing Machine (Instron 3367) by applying load cell of 30000 N with speed of 5 mm min^{-1} after setting the distance between the grip as 50 mm. Tensile strength was determined as 0.24 MPa, breaking at a yield with low strain of 1.4815%. Such feature is a characteristic of a brittle specimen. After yielding, device portrayed constant stress due to presence of PDMS, which is an elastomeric material. Secondly, flexural strength test was conducted using Universal Testing Machine (Instron 34SC-5) by applying load cell of 5000 N with a speed of 0.7 mm min^{-1} after setting the distance between the supports as 25 mm to analyze angle of bend that device can tolerate. Flexural strength was determined as 61.7 MPa, with fracture after bend angle reached 1.213° after loading pin was displaced for 0.8192 mm. Low flexural strength and low bend angle is a characteristic of a brittle material. Similar to the result of the tensile strength, flexural strength and bend angle was greatly influenced by glass. Thirdly, impact resistance test satisfying Izod impact test criteria was conducted using Digital Impact Tester (QM700A, Qmesys) by applying IH1 pendulum as the impact source. The purpose of the pendulum-driven impact test is to measure the energy required to break a standardized specimen satisfying the Izod impact test. Applying 3.7 kJ m^{-2} with pendulum resulted to complete break of the device. For the Izod impact resistance test, such low impact resistance is also characteristic of a brittle material, further indicating that the mechanical properties of the device are governed by the rigid glass substrate.

Additionally, in response to the reviewer's request, it is feasible to fabricate TPIEA in a completely flexible form. The selection of a glass substrate is motivated by its suitability for application on screen panels for image display, taking into account factors such as durability and transparency. The manufacturing process of TPIEA relies on photolithography, enabling its effective production utilizing a polyimide film as a substrate (Supplementary Fig. 38A). Devices manufactured with flexible substrate exhibit superior transparency and flexibility (Supplementary Fig. 38B). Although the angle of bend and impact resistance of TPIEA produced with a flexible substrate were not quantified due to its inherent flexibility, the tensile strength was measured as 54.0 MPa (see Supplementary Fig. 38C). Moreover,

during electrotactile sensation experiments, TPIEA utilizing a flexible substrate demonstrated a comparable electrotactile sensation to that of a glass substrate under medium pressure level (Supplementary Fig. 38D-E). Consequently, TPIEA is possible to construct the device to be completely flexible. These findings have been incorporated into the supplementary materials and the manuscript.

Revised Manuscript (Page 7):

Mechanical properties such as tensile strength, flexural strength, and impact resistance of TPIEA were also examined to verify the stability of the device under repetitive use. Since the majority of the device's volume or area is occupied by the substrate, these mechanical properties are strongly influenced by the materials of the substrate (Supplementary Note 2).

Revised Manuscript (Page 20):

Moreover, the TPIEA can be fabricated on different substrates, making it adaptable for use in flexible display technologies. We have also implemented a flexible version of TPIEA using a PI substrate, demonstrating its adaptability (Supplementary Fig. 38). This versatility in substrate indicates that TPIEA can be integrated into a wide range of display applications, enhancing its potential for future technological innovations⁴⁵.

Revised Manuscript (Page 33):

Also, five healthy adult subjects (aged 25-30; three males and two females) responded to variations in electrotactile sensation due to changes in current density and changes in frequency, using TPEIA on a flexible substrate. The electrical parameters are the same as in the previous experiment.

Revised Manuscript (Page 59):

45. Ren, Z. *et al.* Fully Elastic and Metal-Free Tactile Sensors for Detecting both Normal and Tangential Forces Based on Triboelectric Nanogenerators. *Adv. Funct. Mater.* **28**, 1802989 (2018).

Revised Supplementary Materials (Page 4):

Supplementary Note 2. Mechanical fatigue tests of the TPIEA device

Supplementary Note Fig. 1. Mechanical properties of TPIEA device. (A) Stress-Strain curve and (B) flexural Stress-Strain curve of TPIEA device.

The mechanical characteristics of TPIEA are assessed across three primary classifications, namely tensile strength, flexural strength, and impact resistance. First, tensile strength test using a Universal Testing Machine (Instron 3367) with a 30,000 N load cell at 5 mm min⁻¹ and a 50 mm grip distance determined the tensile strength as 0.24 MPa, with a yield strain of 1.4815%, indicative of a brittle specimen. Post-yield, constant stress was observed due to the elastomeric PDMS. Second, flexural strength test using a Universal Testing Machine (Instron 34SC-5) with a 5,000 N load cell at 0.7 mm min⁻¹ and a 25 mm support distance determined the flexural strength as 61.7 MPa. Fracture occurred at a bend angle of 1.213° with a loading pin displacement of 0.8192 mm, indicative of a brittle material. The flexural strength and bend angle were significantly influenced by the glass substrate, similar to the tensile strength results. Lastly, impact resistance test, conforming to Izod impact test criteria, was conducted using a Digital Impact Tester (QM700A, Qmesys) with an IH1 pendulum. The test, which measures the energy required to break a standardized specimen, showed that applying 3.7 kJ m⁻² resulted in the

complete breakage of the device.

Revised Supplementary Materials (Page 71):

Supplementary Fig. 38. TPIEA device using a flexible polyimide (PI) substrate.

(A) Schematic layouts, and structural designs of the device composed of an electro-tactile actuator array and pressure-sensitive transistor array on PI substrate. (B) Photograph of the transparent electro-tactile actuator with pressure sensor on PI substrate. Scale bar, 2 cm. (C) The tensile strength curve of TPIEA with PI substrate, 54.0 MPa. (D) Electro-tactile intensity for each finger pressure when the current density was changed from 0 to 0.01 A mm⁻². Finger pressure was controlled at 100 kPa. (E) Electro-tactile texture for each finger pressure when the frequency is changed from 0 to 200 Hz.

Comment 5: Although this device is currently utilized solely as a sensor, as indicated, potential applications in diverse fields such as VR, AR, and automotive displays have been suggested. Can the authors elaborate on the feasibility of applying this technology across these varied domains?

Response to Comment 5:

We appreciate the reviewer's comments. Our device is designed to achieve coherent electro-tactile sensation through the simultaneous integration of a pressure sensor and an electro-tactile actuator. Therefore, our study involved a tactile perception experiment illustrated in Figure 3, which was based on the pressure sensor's feedback. This experiment demonstrated the ability to generate diverse tactile sensations by adjusting the frequency, as well as the potential to produce high-resolution tactile images, thereby facilitating innovative tactile communication (Figure 4 to 6). Furthermore, we have outlined potential future applications of this technology across various domains, as highlighted by the reviewer, and have consolidated this information in Supplementary Note 12. Additionally, we have demonstrated how our device, when placed on the back of a subject's hand, can provide tactile feedback consistent with the visual cues presented on a VR device (Supplementary Fig. 47).

Revised Manuscript (Page 25):

The high-resolution of stimulation provided by TPIEA, along with its capacity to modulate a diverse range of tactile sensation, enables its application in various settings, such as integrating visual and tactile components with VR equipment (Supplementary Note 12 and Supplementary Fig. 47).

Revised Manuscript (Page 38-39):

Experiment 12: Electro-tactile sensation implement on VR

One subject (age, 27 years; one female) was instructed to VR device (Meta Quest 2; Oculus Quest) and affix the TPIEA to the dorsal surface of their hand. The texture simulation was implemented to be activated with specific parameters: a current density of 0.03 A mm^{-2} , a frequency of 40 Hz, and a pulse width of 10 ms. Tactile feedback is integrated into the experience by delivering the stimulus prior to virtual contact with the hand during tasks such as bubble touch within the VR environment. TPIEA aimed to enhance the immersive quality of the virtual experience by synchronizing visual and tactile stimuli, pausing the task execution upon sensation detection, and resuming the task when no tactile feedback was detected.

Revised Supplementary Materials (Page 32-33):

Supplementary Note 12. VR and AR application with TPIEA

Virtual reality (VR) and augmented reality (AR) technologies are advancing rapidly across multiple sectors including education, entertainment, and healthcare, offering immersive experiences. While these technologies predominantly rely on visual and auditory stimuli, the provision of tactile feedback has been somewhat limited, often utilizing vibrations from devices like gloves³³. In contrast, electrotactile technology could present a promising solution by delivering a sense of touch through high-resolution and diverse stimulation methods, thereby enhancing the tactile experience in VR environments^{34,35}. An illustrative application involves the integration of electrodes into VR gloves or attaching them to the back of the hand to elicit tactile sensations via electrical stimulation when interacting with virtual objects. Furthermore, the incorporation of electrotactile technology with AR devices facilitates interactions between virtual elements and the physical world. The amalgamation of VR, AR tools, or full-body suits with electrotactile technology holds potential for diverse applications in areas such as education, gaming, medical training, and remote robot manipulation³⁶. To leverage these opportunities effectively, critical considerations include precise control of stimulation, development of personalized feedback systems, implementation of wireless and portable solutions, as well as the commercialization and standardization of these technologies.

Revised Supplementary Materials (Page 80):

Supplementary Fig. 47. Electrotactile sensation application through TPIEA with VR equipment.

Revised Supplementary Materials (Page 85):

Supplementary Video

Supplementary Video 1. Tactile information transmission through TPIEA.

Supplementary Video 2. Display-integrated electrotactile sensation through TPIEA.

Supplementary Video 3. Interference stimulation sensation perception test.

Revised Supplementary Materials (Page 88-89):

33. Hayward, V., Astley, O. R., Cruz-Hernandez, M., Grant, D. & Robles-De-La-Torre, G. Haptic interfaces and devices. *Sens. Rev.* **24**, 16–29 (2004).

34. Strong, R. M. & Troxel, D. E. An Electrotactile Display. *IEEE Trans. Man-Mach. Syst.* **11**, 72–79 (1970).

35. Kourtesis, P., Argelaguet, F., Vizcay, S., Marchal, M. & Pacchierotti, C. Electrotactile Feedback Applications for Hand and Arm Interactions: A Systematic Review, Meta-Analysis, and Future Directions. *IEEE Trans. Haptics* **15**, 479–496 (2022).

36. Shi, Y. *et al.* Self-powered electro-tactile system for virtual tactile experiences. *Sci. Adv.* **7**, eabe2943 (2021).

Comment 6: It is necessary to provide some video materials to show the operation of the devices and the reviewer can evaluate the completeness of the function.

Response to Comment 6:

As the reviewer commented, we provide some supplementary video materials to show the operation of the devices.

Revised Manuscript (Page 18):

It has also shown that adjusting the parameters and locations of electrical stimulation, it can expand the stimulation patterns from a single actuator to multiple actuators, thereby achieving various electrotactile sensations (Supplementary Video 1).

Revised Manuscript (Page 20):

In this way, TPIEA effectively conveyed tactile information corresponding to visual images shown through a display, projecting various tactile information in pictures and videos (Supplementary Fig. 237 and Supplementary Video 2).

Revised Manuscript (Page 22):

This confirms that electrotactile sensation is not realized in bands above 1 kHz, and electrotactile sensation is selectively realized under optimized interference conditions with offset frequencies and not under all interference stimuli (Supplementary Video 3).

Revised Supplementary Materials (Page 85):

Supplementary Video

Supplementary Video 1. Tactile information transmission through TPIEA.

Supplementary Video 2. Display-integrated electrotactile sensation through TPIEA.

Supplementary Video 3. Interference stimulation sensation perception test.

[Response to Reviewer #2]

We thank the reviewer for a thoughtful review and comments about our manuscript, and we welcome the opportunity to address and clarify the issues raised in the reviewer's report. Our responses to the points raised in the report are as follows:

Comment 1: "The manuscript titled "Interference haptic stimulation and consistent quantitative tactility in transparent electrotactile screen with pressure-sensitive transistors" by Lim et al. reports a transparent pressure-calibratable interference electrotactile actuator (TPIEA).

The exploration to use somatosensory evoked potential (SEP) to quantitatively assess the tactile perception and use pressure sensors to calibrate individual pressure variations is interesting and quite effective. Moreover, interference haptic stimulation is utilized to increase the resolution and reduce the current density.

Given the novelty and depth of this work, Reviewer is in support of its publication in Nature communications after major revisions addressing the following:"

"In Fig. 1g, the pressure sensitivity should be characterized under more pressure stages to ensure its accuracy. Moreover, the reason why there are two pressure sensitivities should be clarified."

Response to Comment 1:

We would like to thank the reviewer for bringing this point to our attention. Following the reviewer's suggestion, we have revised Fig. 1g to include data points with more pressure stages, ranging from 5 kPa to 300 kPa with a step size of 15 kPa (Supplementary Fig. 7).

Additionally, to clearly explain the presence of two pressure sensitivities in our pressure sensing system, we have added a plot showing the relationship between the thickness of the dielectric layer and the change in drain current (Supplementary Fig.17). The key mechanism for pressure detection in our pressure-sensitive transistor is utilizing the change in the drain current due to the thickness modulation of a dielectric layer, which consists of a 50 μm -thick polydimethylsiloxane (PDMS) elastomer containing air holes.

In general, the thin film elastomers exhibit a high effective compressive modulus, which changes non-linearly under the applied stress [*Microelectron. Eng.*, **88** (8), 1982-1985 (2011); *AIP Conf. Proc.*, **2272**, 040007 (2020); *Opt. Mater.*, **85**, 153-161 (2018)]. The thickness of the materials decreases rapidly under low compressive force, but the rate of decrease slows at a specific thickness level. In our additional plot, the slope of the PDMS thickness change shifted at an applied pressure of 75 kPa. This behavior correlated with the change in drain current, showing a qualitative dependency on the applied pressure. The sensitivity of our pressure sensor to applied pressure was determined by the linear fit of the plot, which revealed two distinct linear dependencies across different pressure levels.

Revised Manuscript (Page 8):

Furthermore, to examine the sensitivity of pressure-sensitive FET, real-time detection of the change in drain current was conducted during applying the external compressive pressure from 5 kPa to ~~300250~~ kPa (Fig. 1f and Supplementary Fig. 7). The sensitivity was calculated as $7.46 \times 10^{-3} \text{ kPa}^{-1}$ ~~0.0067 kPa^{-1}~~ within the pressure range from 1 to 75 kPa, and 1.22×10^{-3} ~~0.0017 kPa^{-1}~~ within the pressure range from 75 to ~~300250~~ kPa (Fig. 1g).

Revised Manuscript (Page 12):

The vertical contraction of PDMS increased the capacitance between the gate and channel, which led to an increase in drain current, and the relative change in drain current was used to measure the touch pressure (Supplementary Fig. 17).

Revised Manuscript (Page 41-42):

Fig. 1: Transparent pressure-calibratable interference electroactuator

Revised Manuscript (Page 42):

f, Real-time detection of relative change in drain current of the pressure-sensitive transistor of TPIEA at pressure ranges from 5 kPa to ~~250~~75 kPa. **g**, Relative changes in drain current of the pressure-sensitive transistor of TPIEA by the applied pressure at the ranges from 5 kPa to ~~250~~300 kPa, and the pressure sensitivity of pressure sensor (red line).

Revised Supplementary Materials (Page 40):

Supplementary Fig. 7. Real-time detection of relative changes in drain current at (A) low-pressure ranges (<75 kPa), (B) and high-pressure ranges (>75 kPa, <300 kPa). Each step corresponds to 5, 15, 30, 45, 60, 75, 90, 105, 120, 135, 150, 165, 180, 195, 210, 225, 240, 255, 270, 285, and 300 kPa.

Revised Supplementary Materials (Page 50):

Supplementary Fig. 17. Thickness of an air-dielectric layer (red line) and relative changes in drain current (black) under applied pressure.

Comment 2: “In Fig. 3f, only 100 kPa is adopted in the examination. It’s better to try more pressures to verify the importance of pressure control.”

Response to Comment 2:

We appreciate the reviewer’s valuable comment. As the reviewer suggested, we have examined the subjects’ perception responses when their finger pressure was set at 40 kPa and 170 kPa.

Revised Manuscript (Page 14-15):

Perception responses at finger pressures of 40 kPa and 170 kPa are also presented in Supplementary Fig. 21. ~~The difference in the accuracy is clearly in Fig. 3g and 3h.~~ The distribution of perception at stimulation level 3 is widely spread before pressure control, whereas the distribution is narrowly concentrated at ~~100 kPa of~~ controlled pressure (Figs. 3g, 3h and Supplementary Fig. 22).

Revised Supplementary Materials (Page 54):

Supplementary Fig. 21. Perception response to the stimulation composed of six levels of current amplitude and confusion matrix (inset) with (A) 40 kPa and (B) 170 kPa (B) of controlled pressure.

Revised Supplementary Materials (Page 55):

Supplementary Fig. 22. Distribution of responses at a specific stimulation level with a controlled pressure of (A) 40 kPa (B) and 100 kPa.

Comment 3: Besides pressure, skin wetness is also one of the important parameters that influences the electrotactile sensation. Maybe there should be some discussions here.

Response to Comment 3:

We appreciate the reviewer’s insightful questions. As the reviewer suggested, it is crucial to investigate how skin wetness can affect electrotactile sensation. Therefore, we included a skin wetness perception test to address concerns regarding humidity.

Specifically, skin moisture levels can enhance skin hydration and conductivity, consequently altering the conductivity of the stratum corneum dielectric layer [Measurement, **62**, 215-221 (2015); 2007 IEEE 33rd Annual Northeast Bioengineering Conference, Stony Brook, NY, USA, 2007, pp. 110-111; IEEE Trans. on Biomed. Eng., **63** (2), 423-430 (2016); Biosensors, **8** (2), 31 (2018)]. A decrease in humidity leads to an increase in skin-electrode impedance, while an increase in humidity results in the opposite effect. Since the skin-electrode impedance changes not only with the pressure but also the humidity of skin, it is important to control the humidity for consistent results.

In our study, the management of humidity can be categorized into three primary classifications. These encompass the humidity levels within the device, external to the device, and the humidity of the skin. Initially, the internal humidity of the device was regulated during the manufacturing process by creating an air-gap in a vacuum state and sealing it to safeguard the pressure-sensitive component from the effects of moisture. Moreover, external humidity was controlled at a consistent level by utilizing a temperature and humidity control unit (Supplementary Note Fig. 10A). The temperature and humidity control unit in our experiment operates on the principle of regulating both temperature and humidity within a closed environment to predetermined levels. This is achieved through a combination of sensors, cooling/heating elements, and a humidifier/dehumidifier system. The sensors continuously monitor the environmental conditions, and the control system adjusts the heating or cooling elements to maintain the desired temperature. Similarly, the humidifier adds moisture to the air when the humidity drops below the set level, while the dehumidifier removes excess moisture when the humidity exceeds the desired level. Furthermore, in our experiment, the temperature was consistently maintained at 26°C. The humidity was controlled and tested under 43.1%. Subsequently, as we focused on typical skin conditions, all trials were conducted while maintaining skin humidity within the range of 38-42%. Furthermore, after a break, the skin humidity was maintained at 38-42% after sterilization and drying before proceeding with the experiment. This precise control ensured that the external environment of the device remained stable, providing reliable and consistent conditions for the experiment. The careful management of both temperature and humidity is crucial for ensuring the accuracy and repeatability of our results.

To further investigate the perception of electrotactile sensation in relation to skin humidity levels, experiments were conducted to assess how this sensation is experienced under varying skin moisture conditions. These experiments were designed to explore the impact of skin humidity on the intensity of electrotactile sensation, which shares similarities with finger pressure. Skin humidity levels were categorized into four groups: dry (30-36%), normal (36-42%), hydrated (43-50%), and very hydrated (50-60%) states (Supplementary Note Fig. 10B). Prior to the experiments, subjects measured the humidity levels of their skin and then proceeded with the electrotactile intensity experiment

following the same method as in Experiment 3. Throughout the experiment, pressure was initially standardized at a medium level of 100 kPa.

It indicated that an increase in skin moisture levels resulted in enhanced electro tactile perception at lower current densities. The threshold current density required to reach the "Perceptible" level showed no significant variation across dry, normal, and hydrated skin conditions at 0.0125 A mm^{-2} . However, with higher current densities, the thresholds for reaching the "Very well perceptible" and "Pain" levels gradually decreased. Notably, in the highly hydrated condition, all stages, including the "Perceptible" level, exhibited lower thresholds compared to the normal condition. The pain stage threshold was the lowest at 0.0575 A mm^{-2} (Supplementary Note Fig. 10C-F). Humidity is a factor that impacts skin-electrode impedance. Therefore, our research involved conducting experiments under controlled humidity conditions (38-42%). Throughout our cognitive experiments and applications, we avoided scenarios where electronic devices were operated with wet hands (humidity exceeding 50%) to maintain standard conditions.

The efficacy of managing skin-electrode impedance in future electro tactile technology is crucial. Therefore, in addition to the pressure correction methods detailed in our study, efforts should be made to develop additional electro tactile device capable of controlling skin humidity. These contents have been included in the manuscript and supplementary materials.

Revised Manuscript (Page 15):

In addition, the skin-electrode impedance changes with not only the pressure but also the humidity of skin, it is important to control the humidity for consistent results. Therefore, another factor that affects electro tactile sensation, skin humidity, was also controlled and adjusted to maintain consistent conditions throughout the tests under varying skin moisture conditions (Supplementary Note 9).

Revised Manuscript (Page 31-32):

The experiments with human subjects were performed in compliance with all the ethical regulations under a protocol that was approved by the Yonsei University Institutional Review Board (application no. 7001988-202303-HR-1809-02) via a haptic interface with no additional human-subject risk, following the provided study guidelines. The subjects in all experiments were individuals of both genders, between the ages of 22 and 40. Given that display and device usage are not restricted by gender or sex, we included both male and female subjects in our recruitment process. All ~~of the~~ subjects ~~provided-gave~~ written informed consent about the experimental procedure. All subjects were trained to manipulate the electro tactile system with the help of experimenters until they ~~were familiarized with~~ ~~understood~~ the sensation of electrical stimulation. Subjects were asked to clean their fingers with alcohol to minimize effects of moisture before the test, ~~maintaining a 36-42% skin humidity~~. For all tests, subjects underwent a 5- to 10- min familiarization phase to acclimatize themselves with the task. ~~All of the subjects prepared for dull sensations by having a 5-minute break time after conducting a 10-minute cognitive experiment to prevent the reduction in tactile perception. Also, even when they restarted, they sanitized their hands with alcohol and wiped them off to maintain a 36-42% humidity level on the skin surface. A temperature and humidity control unit was utilized to maintain a consistent level of humidity for controlled environment: 26°C and under 43.1% humidity. Also, all cognitive experiments were conducted under the advice and observation of the neurosurgeon.~~

Revised Manuscript (Page 39):

Experiment 13: Electro tactile sensation in a varying humidity condition

Skin humidity levels were categorized into four groups: dry (30-36%), normal (36-42%), hydrated (43-50%), and very hydrated (50-60%) states. Prior to the experiments, subjects measured the humidity levels of their skin by skin analyzer (RoHS compliant; China) and then proceeded electro tactile intensity experiment following the same method as in Experiment 3. Throughout the experiment, pressure was initially standardized to a medium level of 100 kPa.

Revised Supplementary Materials (Page 25-27):

Supplementary Note 9. Electro tactile sensation in different humidity conditions

Supplementary Note Fig. 10. Experiment on electrotactile sensation in a varying humidity setting. **(A)** Photograph of a temperature and humidity control unit. A temperature and humidity control unit was utilized to maintain a consistent level of humidity for controlled environment: 26°C and under 43.1% humidity. **(B)** Table categorizing skin conditions according to skin humidity. Hygrometer readings and corresponding electrotactile intensity for each skin condition on **(C)** dry condition, **(D)** normal condition, **(E)** hydrated condition, and **(F)** very hydrated condition. All scale bars = 3 cm.

Skin humidity is a significant factor affecting the electrotactile sensation, as it can impact skin hydration and conductivity. Specifically, skin moisture levels can enhance skin hydration and conductivity, consequently altering the conductivity of the stratum corneum layer²²⁻²⁵. A decrease in humidity leads to an increase in skin-electrode impedance, while an increase in humidity results in the opposite effect. Given that skin-electrode impedance is influenced by both pressure and skin humidity, we also focused on measuring the

variations in electrical sensation corresponding to different humidity levels.

The laboratory maintained temperature and humidity levels using a temperature and humidity control unit (Supplementary Note Fig. 10A). The temperature and humidity control unit in our experiment operates on the principle of regulating both temperature and humidity within a closed environment to predetermined levels. This is achieved through a combination of sensors, cooling/heating elements, and a humidifier/dehumidifier system. The sensors continuously monitor the environmental conditions, and the control system adjusts the heating or cooling elements to maintain the desired temperature. Similarly, the humidifier adds moisture to the air when the humidity drops below the set level, while the dehumidifier removes excess moisture when the humidity exceeds the desired level. Humidity regulation within the device was achieved by integrating and encapsulating the system in a vacuum environment with an air-gap.

Human skin condition was categorized into dry (30-36%), normal (36-42%), hydrated (43-50%), and very hydrated (50-60%) based on the humidity levels (Supplementary Note Fig. 10B)²⁶. Prior to the experiments, subjects measured the humidity levels of their skin by skin analyzer (RoHS compliant; China) and then proceeded electro tactile intensity experiment following the same method as in Experiment 3, with finger pressure initially set to a standardized medium pressure of 100 kPa. It indicated that an increase in skin moisture levels led to a heightened electro tactile perception at lower current densities. The threshold current density required to reach the "Perceptible" level showed no significant variation across dry, normal, and hydrated skin conditions at 0.0125A mm⁻². However, with higher current densities, the thresholds for reaching the "Very well perceptible" and "Pain" levels gradually decreased. Notably, in the highly hydrated condition, all stages, including the "Perceptible" level, exhibited lower thresholds compared to the normal condition, with the pain stage threshold being the lowest at 0.0575 A mm⁻² (Supplementary Note Fig. 10C-F). This investigation aimed to explore the impact of skin humidity on electro tactile sensation, with results confirming that skin humidity can influence the perception of electro tactile stimulation. It is important to note that all experiments, except for this study, focused on normal skin conditions with a maintained humidity level of 38-42%. Additionally, conditions involving cognitive experiments, such as handling electronic devices with wet hands (humidity exceeding 50%), were not considered.

Revised Supplementary Materials (Page 87-88):

22. Lu, F. *et al.* Review of Stratum Corneum Impedance Measurement in Non-Invasive Penetration Application. *Biosensors* **8**, 31 (2018).
23. Yokus, M. A. & Jur, J. S. Fabric-Based Wearable Dry Electrodes for Body Surface Biopotential Recording. *IEEE Trans. Biomed. Eng.* **63**, 423–430 (2016).
24. Jayaraman, A., Kaczmarek, K. A., Tyler, M. E. & Okpara, U. O. Effect of localized ambient humidity on electro tactile skin resistance. *2007 IEEE 33rd Annual Northeast Bioengineering Conference* 110–111 (2007).
25. Li, D. *et al.* Non-invasive measurement of normal skin impedance for determining the volume of the transdermally extracted interstitial fluid. *Measurement* **62**, 215–221 (2015).

Comment 4: The influence of current density and frequency on electro tactile sensation has been extensively evaluated in this work. So how about pulse width? How does the pulse width affect the electro tactile sensation? Moreover, in Supplementary Fig. 16, the dots and dashes in the electrical simulation are distinguished by different pulse widths (10 ms and 500 ms). But it seems that the same 10 ms pulse width with different pulse numbers (e.g., 1 pulse and sequential 20 pulses) can achieve the same results.

Response to Comment 4:

We are grateful to the reviewer for valuable comment. Pulse width denotes duration for which the electrical stimulus is active, delineated into periods of activation (on time) and deactivation (off time). Modulating the duty cycle percentage of the pulse wave results in alterations to the pulse width (Supplementary Fig. 18). By modifying the pulse width of electro tactile, it is possible to elicit two significant alterations in tactile perception. Initially, an increase in pulse width enables the generation of a sustained pressure and an elongated pulling sensation, as the stimulation can be consistently applied, in contrast to shorter pulse widths. Secondly, as illustrated in Supplementary Figure 18, manipulation of the duty cycle can serve as a method for individuals to distinguish between 0 and 1, facilitating binary information transmission. It is a well-established practice to utilize pulse width modulation for the purpose of information transfer [Pulse width modulation for power converters: principles and practice, **18**, (1-744), 2003, John Wiley & Sons; Dynamics and Control of Switched Electronic Systems: Advanced perspectives for modeling,

simulation and control of power converters, *Pulse-Width Modulation*, 25-61, 2012, Springer]

First, in order to investigate the changes in electrotactile sensation according to pulse width, we varied the pulse width to 10 ms, 100 ms, 300 ms, 500 ms, 700 ms, and 900 ms at a frequency of 1 Hz and a current density of 0.015 A mm^{-2} , and asked the subjects to respond with "Brief touch," "Pulling sensation," "Sustained pressure," or "Very long stimulus." In the response results, it was observed that at a latency of 10 milliseconds, all responses were categorized as "Brief touch." However, there was a shift from "Brief touch" at 100 milliseconds, and with the increment in pulse width, there was an increased tendency to transition from a "Pulling sensation" to "Sustained pressure." Furthermore, starting from 500 milliseconds, there was an increase in the ratio of the "Very long stimulus." Consequently, it was ascertained that a longer pulse width has the capacity to induce a persistent sensation (Supplementary Fig. 19).

Second, to validate the sensation variation based on variations in pulse width, the experiment corresponding to Supplementary Fig. 26 was conducted. Dots representing 0 were transmitted at a pulse width of 10 ms (1% duty cycle), while dashes representing 1 were transmitted at a pulse width of 500 ms (50% duty cycle) to convey Morse code information. As noted by the reviewer, it is possible to distinguish between dots and dashes even with varying pulse numbers (e.g., 1 pulse and sequential 20 pulses) when using a pulse width of 10 ms. However, if the pulse number decreases or the interval changes, leading to an increase in the off time, subjects may misinterpret it as a high-frequency stimulation rather than sequential stimulation. Therefore, adjusting the pulse width enables clearer transmission of information (Figure R3).

Figure R3. Compare pulse waves with a pulse width of 10 ms with varying numbers of sequential pulses with a pulse wave with a pulse width of 500 ms

The manipulation of pulse width demonstrates a significant correlation with frequency. Broadening the pulse width limits the increase in frequency, therefore, a wide pulse width is feasible within the low frequency spectrum. Unlike the persistent perception elicited by pulse width, frequency is linked to diverse texture sensations. Consequently, applications have focused on introducing various texture sensations through frequency modulation, after establishing a fixed pulse width of 10 ms. Nevertheless, it was revealed that by modifying pulse width within the low frequency spectrum, a novel persistent sensation could be induced. The TPIEA device demonstrates the capacity to generate

various tactile sensation and to communicate tactile information by regulating pulse width, frequency, and amplitude in electro-tactile stimulation. These contents have been integrated into the manuscript and supplementary materials.

Revised Manuscript (Page 13-14):

The pulse width was also independent of pressure. Pulse width denotes the length of time for which the electrical stimulus is active, delineated into periods of activation (on time) and deactivation (off time) (Supplementary Fig. 18). By modifying the pulse width of electro-tactile, it is possible to elicit two significant alterations in tactile perception. Initially, an increase in pulse width enables the generation of a sustained pressure and an elongated pulling sensation, as the stimulation can be consistently applied, in contrast to shorter pulse widths (Supplementary Fig. 19). Secondly, manipulation of the duty cycle can serve as a method for individuals to differentiate between 0 and 1, facilitating binary information transmission. It was subsequently verified through Morse code within the application.

Revised Manuscript (Page 33):

In order to investigate the changes in electro-tactile sensation according to pulse width, we varied the pulse width to 10 ms, 100 ms, 300 ms, 500 ms, 700 ms, and 900 ms at a frequency of 1 Hz and a current density of 0.015 A mm⁻², and asked the subjects to respond with "Brief touch," "Pulling sensation," "Sustained pressure," or "Very long stimulus." The pulse width test was repeated five times for each subject. Also, five healthy adult subjects (aged 25-30; three males and two females) responded to variations in electro-tactile sensation due to changes in current density and changes in frequency, using TPEIA on a flexible substrate. The electrical parameters are the same as in the previous experiment.

Revised Supplementary Materials (Page 51):

Supplementary Fig. 18. Modulation of pulse width determined by the duty cycle.

Revised Supplementary Materials (Page 52):

Supplementary Fig. 19. The tendency of the response when the pulse width is 10 ms, 100 ms, 300 ms, 500 ms, 700 ms, and 900 ms. The subjects were encouraged to respond with "Brief touch," "Pulling sensation," "Sustained pressure," or "Very long stimulus." In the response results, it was observed that at a latency of 10 milliseconds, all responses were categorized as "Brief touch." However, there was a shift from "Brief touch" at 100 milliseconds, and with the increment in pulse width, there was an increased tendency to transition from a "Pulling sensation" to "Sustained pressure." Furthermore, starting from 500 milliseconds, there was an increase in the ratio of the "Very long stimulus." Consequently, it was ascertained that a longer pulse width has the capacity to induce a persistent sensation.

Comment 5: Interference haptic stimulation is utilized to increase the resolution in this work. There is also a super-resolution strategy in "Science advances 8 (36), eabp8738". Please compare these two strategies and have a discussion.

Response to Comment 5:

We are very grateful for the reviewer's insightful comment. As the reviewer mentioned, we compared our interference haptic stimulation with the super-resolution strategy of [*Sci. adv.* **8** (36), eabp8738 (2022)], and summarized the advantages of interference haptic stimulation using two tables. Also, we added specific mathematical theorems and calculation methods of interference haptic stimulation.

The super-resolution technique described in the publication [*Sci. adv.* **8** (36), eabp8738 (2022)] involves a single, non-interfering stimulation and only alters the arrangement of anodes and cathodes. In particular, the approach maintains a constant number of anodes while varying the quantity and layout of cathodes to manipulate the stimulated location, thereby demonstrating a super-resolution strategy.

In contrast, the interference haptic stimulation of our work represents a distinct form of electrical stimulation from single stimulation. This method requires two pairs of anode and cathode. By the interference phenomenon resulting from the superposition of two electric fields, stimulation can be carried out at locations with a high modulation index, allowing for a significant decrease in current density compared to single stimulation using only one pair of anode and cathode. Additionally, by utilizing interference waves when the amplitudes of each electric field are equal, stimulation can be achieved at the center positions of two pairs, enabling stimulation at localized points through interference even if the electrode spacing is not ultra-high resolution.

In brief, our work introduced a novel haptic stimulation technique known as interference haptic stimulation. This method enabled a significant reduction in current density and more precise stimulation during interference haptic stimulation. In order to verify this, we have attached a table comparing our work with [*Sci. adv.* **8** (36), eabp8738 (2022)] in terms of electrode conditions, stimulus conditions, stimulus resolution, and other aspects (Table R3).

		Our work	Science advances 8 (36), eabp8738 (2022)
Electrode condition	Electrode diameter	0.5 mm	2 mm
	Electrode area	0.196 mm ²	3.14 mm ²
	Electrode distance	2 mm	4 mm
Stimulation condition	Stimulation type	2 (Single stimulation + Interference stimulation)	1 (Single stimulation)
	Type of stimulation site	Normal site, High-resolution Normal site, Center site, Interference site	Normal site, Super-resolution site
	Stimulation current density	8.5 A/mm ² (Single stimulation) 5.6 A/mm ² (Interference stimulation)	10 A/mm ²
Stimulation resolution	Number of electrodes in finger area (Finger area: 8 mm × 8 mm)	25	9
	Number of normal stimulation sites in finger area	25	9
	Number of high-resolution stimulation sites in finger area	136 (80 + 40 + 16)	23
	Number of total stimulation site in finger area	161	32
	Tactile spatial resolution in finger area	2.51 Points/mm ²	0.5 Points/mm ²
	Number of total simulation sites for 9 electrodes	49 (9+40)	32 (9+23)
etc	Number of tactile textures	9	5
	Transmittance	89%	0%
	Pressure calibration	YES	NO

Table R3. Comparison of electrotactile strategy in our work and [*Science advances* 8 (36), eabp8738 (2022)]

Our study utilizes electrodes that have a spacing distance of 2 mm, produced through the photolithography technique. These electrodes are designed to meet the requirements for human two-point discrimination. Additionally, our research involves the application of two types of electrical stimulation: single stimulation and interference stimulation.

To assess potential changes in current density resulting from Interference stimulation, we have re-calculated current density to match the simulated condition in [*Sci. adv.* 8 (36), eabp8738 (2022)]. The driving current density during interference stimulation was 5.6 A/mm², representing a 44% reduction compared to the 10 A/mm² reported in [*Sci. adv.* 8 (36), eabp8738 (2022)].

Moreover, a comparative analysis was undertaken between our work and [*Sci. adv.* 8 (36), eabp8738 (2022)], focusing on resolution within a consistent area size. Assuming a finger area of 8 mm × 8 mm, our work identified a total of 161 stimulation sites, while [*Sci. adv.* 8 (36), eabp8738 (2022)] contained 32 stimulation sites. The tactile spatial resolution was calculated to be 2.51 Points/mm² for our work and 0.5 Points/mm² for [*Sci. adv.* 8 (36), eabp8738 (2022)], indicating a resolution that is approximately 400% higher in our study (Figure R4).

Figure R4. Schematic illustration of the distribution of various electro tactile sites between studies our work and [*Sci. adv.* 8 (36), eabp8738 (2022)] in 8 mm × 8 mm finger area.

When the number of electrodes was adjusted to nine, our study observed a total of 49 stimulation sites, compared to 32 in [*Sci. adv.* 8 (36), eabp8738 (2022)] (Figure R5).

Figure R5. Schematic illustration of the distribution of various electro-tactile sites between studies our work and [*Sci. adv.* **8** (36), eabp8738 (2022)] with 9 electrodes.

This supports that interference haptic stimulation allows for electrical haptic stimulation at reduced current densities and with enhanced resolution compared to single stimulation. Also, other number of tactile textures, transmittance, pressure calibration are also compared in the table R3.

Furthermore, we tabulate the advantages (current density and high-resolution density) of our work's interference haptic stimulation compared to other electro-tactile prior studies, including [*Sci. adv.* **8** (36), eabp8738 (2022)]. The current densities were calculated using the skin impedance in the 1 kHz [*IEEE Trans. Biomed. Eng.*, **35** (8), 649-651 (1988)] band and the device resistance of each study for consistency of condition (Supplementary Table 3). The results of this comparison have been summarized and added to the manuscript and Supplementary Fig. 42.

Also, to aid in the comprehension of the operational principles of interference haptic stimulation and the associated computations for modulation index, frequency, and amplitude, an explanation has been provided in the Supplementary Note 11.

Revised Manuscript (Page 20):

Interference haptic stimulation exploits the interference between two high-frequency electric fields with offset frequencies to achieve electro-tactile stimulation at a deep skin position beyond the electrode surface through the generation of interference waves (Supplementary Note 611).

Revised Manuscript (Page 24):

The superiority of spatial resolution and low current density of the interference stimulation was demonstrated in Supplementary Fig. 42 and Supplementary Table 3.

Revised Supplementary Materials (Page 28-32):

Interference stimulation operates based on two assumptions: the interference phenomenon caused by the superposition of the low-pass filtering property and the electric field. First, the Low-pass filtering property is a theory that the neural membrane does not consciously respond to electrical fields above 1 kHz³⁰. Next, let's call the altering current generated by a pair of anode and cathode as I₁, and the altering current generated by another pair of anode and cathode as I₂. Due to the interference phenomenon where the electric fields created by these two currents overlap, a new electric field can appear^{31,32}. Simply put, when the frequency of I₁, f₁, is above 1 kHz with an amplitude of Z, and the frequency of I₂, f₂, is f₂ = f₁ + Δf (Δf ≤ 200 Hz) with an amplitude of Z, due to the superposition of the two electric fields, an offset frequency of Δf is generated. Theoretically, an altering modulation electric field with a maximum amplitude of 2Z is created when it is perfect interference condition. This is based on the formation of beating (acoustic) through superposition when two overlapping electric fields have different frequencies, resulting in the creation of a wave in the form of an envelope. The generated wave is referred to as an envelope wave.

The electric field generated by the interference phenomenon caused by the superposition of I₁ and I₂ approached mathematically is expressed as follows.

a₁ = wave 1

a₂ = wave 2

a = envelop wave

t = time

ω_n = frequency of wave n

ω = frequency of envelop wave

A_n = amplitude of wave n

A = amplitude of envelop wave

A_{max} = maximum amplitude of envelop wave

A_{min} = minimum amplitude of envelop wave

μ = modulation index value

E_{AM} = envelope modulation amplitude

$$A_1 \sin \omega_1 t = a_1 \dots (1)$$

$$A_2 \sin \omega_2 t = a_2 \dots (2)$$

$$A \sin \omega t = a \dots (3)$$

To calculate A, start with A₂ as the reference and add the amplitude change of Wave 1.

$$A = A_2 + a_1 \dots (4)$$

$$A = A_2 + A_1 \sin \omega_1 t \dots (5)$$

Accordingly, A_{max} is obtained when sin ω₁t is +1, and A_{min} is obtained when sin ω₁t is -1.

$$A_{\max} = |A_1 + A_2| \dots (6)$$

$$A_{\min} = |A_1 - A_2| \dots (7)$$

Add equation 6 and equation 7.

$$A_{\max} + A_{\min} = 2A_2$$

$$A_2 = A_{\max} + A_{\min} / 2 \dots (8)$$

Subtract equation 7 from equation 6

$$A_{\max} - A_{\min} = 2A_1$$

$$A_1 = A_{\max} - A_{\min} / 2 \dots (9)$$

$$A_1 / A_2 = A_{\max} - A_{\min} / A_{\max} + A_{\min} = \mu \dots (10)$$

When perfect modulation occurs, μ has a value of 1.

Also, according to the principle of beating, ω is as follows.
 $\omega = \omega_1 - \omega_2$ (beating frequency) = offset frequency... (11)

Therefore, equation 3 can be summarized as follows.
 $a = A_2(1 + \mu \sin \omega_1 t) \sin (\omega_1 - \omega_2) t \dots$ (12)

The ideal value of μ is 1. However, due to factors such as the attenuation of amplitude during the propagation of the electric field through the medium and the phase discrepancies among waves, μ typically assumes a value of 1 or less. The difference between the maximum (A_{\max}) and minimum (A_{\min}) amplitudes during this process is termed the envelope modulation amplitude (E_{AM}), which signifies the strength of the actual stimulation within the envelope wave. E_{AM} varies based on the magnitude of μ , increasing with higher values of μ .

To achieve a high μ , it is necessary for the amplitudes A_1 and A_2 at the point of interference to be similar. E_{AM} is maximized at the location where 'the values of A_1 and A_2 interfere most similarly, leading to stimulation, while at other locations, the E_{AM} does not have enough values to surpassing the threshold, preventing stimulation. Therefore, Interference stimulation can be described as a stimulation method using the phenomenon of interference, and to summarize:

1. Stimulation occurs at the position where the μ is maximized. When A_1 and A_2 are the same, the center position is the position where μ is maximized. By adjusting the position of μ , stimulation can be carried out at different locations.
2. The intensity of the stimulation is E_{AM} , so theoretically it can deliver a stronger stimulus than A_1 and A_2 .
3. The frequency of stimulation is the offset frequency, which is the difference between the two waves.

Revised Supplementary Materials (Page 75):

Supplementary Fig. 42. Comparisons of our work with previous studies (A) for tactile stimulation density, (B) number of total stimulation sites for 9 electrodes, and (C) stimulation current density. Blue star is single stimulation and red star is interference stimulation in our work.

Revised Supplementary Materials (Page 83):

	Tactile stimulation resolution density (points mm⁻²)	Number of total stimulation sites for 9 electrodes	Stimulation current density (A mm⁻²)		Ref
This work	2.37	49	8.5	5.6	-
Science Advances 8 (36), eabp8738 (2022).	0.5	32	10		⁴⁵
Science Advances , 7 (6), eabe2943 (2021).	0.0025	9	9.5		³⁶
IEEE Transactions on Haptics , 5 (2), 184-188 (2012).	1.77	9	10		⁴⁶
IEEE Transactions on Biomedical Engineering , 46 (8), 929-936 (1999).	0.11	9	13.3		⁴⁷
ACS omega , 3 (1), 662-666 (2018).	0.05	13	10		⁴⁸

Supplementary Table 3. Comparison of the tactile resolution and stimulation current density of our work with other previous studies.

Revised Supplementary Materials (Page 88):

30. Hutcheon, B. *et al.* Resonance, oscillation and the intrinsic frequency preferences of neurons. *Trends Neurosci.* **23**, 216–222 (2000).
31. Mirzakhali, E., Barra, B., Capogrosso, M. & Lempka, S. F. Biophysics of Temporal Interference Stimulation. *Cell Syst.* **11**, 557-572 (2020).
32. Grossman, N. *et al.* Noninvasive Deep Brain Stimulation via Temporally Interfering Electric Fields. *Cell* **169**, 1029-1041 (2017).

Comment 6: In Fig. 6d, interference haptic stimulation is utilized to increase the recognition accuracy among four patterns. But these four patterns are totally different from each other. It is suggested to further evaluate the high-resolution strategy by distinguishing patterns with similar configurations such as curves with slightly different angles and ellipses with small changes in aspect ratios.

Response to Comment 6:

We are very grateful for the reviewer’s insightful comment. As the reviewer suggested, to illustrate the potential of interference haptic stimulation in discerning patterns with similar configurations through higher resolution stimulation, two supplementary experiments were carried out.

The initial experiment involved subjects matching the number of points in flames when the number of points in flames were adjusted. The flames consisted of 3, 4, and 5 points, with distinct arrangements enabling differentiation through high-resolution patterns, but the large form is the same. This experiment, involving ten subjects (aged 25-35 years; five males and five females) with occluded vision, entailed testing each point quantity five times in random order per subject, repeated 5 times. The results indicated an accuracy rate of 93.4%, demonstrating the ability to distinguish the number of points in flames with a high degree of certainty using high-resolution stimulation patterns (Supplementary Fig. 45).

Furthermore, a subsequent experiment focused on matching the direction of flames. Following the preparation of flames with an identical number of points, the two primary flame directions (right, left) were distinguished. They were categorized into four groups based on the orientation of the main flame points: (left, left, right) (right, right, right) (left, left, left) (left, right, right). The cognitive experiment mirrored the methodology of the preliminary study, with each subject repeating it 5 times. The findings revealed an accuracy rate of 92.5%, affirming that interference stimulation can effectively differentiate even subtly different directions of the same flame through high-resolution tactile implementation (Supplementary Fig. 46).

During the experiments, single stimulation was set at an intensity of 0.0175 A mm^{-2} , while interference stimulation was at 0.01 A mm^{-2} , as depicted in Figure 6. The outcomes of both experiments were depicted in confusion matrices, with comprehensive details and methodologies of the experiments provided in the main text and supplementary materials.

Revised Manuscript (Page 25):

This haptic stimulation also demonstrated the capability to differentiate patterns with similar configurations, such as fire patterns with slightly different orientations or numbers of flames (Supplementary Fig. 45 and 46).

Revised Manuscript (Page 37):

Experiment 8: High-resolution assessment on interference electrotactile sensation

Ten healthy adult subjects (age, 25–35 years; five males and five females) were arranged in the recognition test to illustrate the potential of interference haptic stimulation in discerning patterns with similar configurations through higher resolution stimulation. The initial experiment involved subjects matching the number of points in flames when the number of points in flames were adjusted. The flames consisted of 3, 4, and 5 points, with distinct arrangements enabling differentiation through high-resolution patterns, but the large form is the same. Subjects with occluded vision, entailed testing each point quantity five times in random order per subject, repeated ten times. Next, a subsequent experiment focused on matching the direction of flames. Following the preparation of flames with an identical number of points in flame, the two primary flame directions (right, left) were distinguished. They were categorized into four groups based on the orientation of the main flame points: (left, left, right) (right, right, right) (left, left, left) (left, right, right). The cognitive experiment mirrored the methodology of the preliminary study, with each subject repeating it five times.

Revised Supplementary Materials (Page 78):

Supplementary Fig. 45. High-resolution interference tactile stimulation that can distinguish between different points in flame patterns.

(A) Three different flame patterns with different numbers of prepared points: 5 points, 4 points, 3 points. **(B)** Schematic images of the complex pattern and stimulation sequence of a flame created by combining single stimulation sites with interference stimulation sites. The number of points in the flame is 5, 4, and 3 from left to right. **(C)** Confusion matrix results of 50 perception trials corresponding to each pattern. The results indicated an accuracy rate of 93.4%.

Revised Supplementary Materials (Page 79):

Supplementary Fig. 46. High-resolution interference tactile stimulation that can distinguish between different directions of flames.

(A) Four different patterns of the main fireworks directions: (left, left, right), (right, right, right), (left, left, left), (left, right, right). **(B)** Schematic images of the complex pattern and stimulation sequence of a flame created by combining single stimulation sites with interference stimulation sites which is distinguished in directions of flame. **(C)** Confusion matrix results of 50 perception trials corresponding to each pattern. The findings revealed an accuracy rate of 92.5%.

Comment 7: A suggestion to reorganize the figures. The aspect ratio of the current figures seems too large.

Response to Comment 7:

We appreciate the reviewer's helpful advice. As the reviewer suggested, we have replaced the large aspect ratio figures 1 and 3 with adjusted ones.

Revised Manuscript (Page 41-42):

Fig. 1: Transparent pressure-calibratable interference electroactuator

Revised Manuscript (Page 45-46):

Fig. 3: Experimental result of tactile perception test with pressure control

[Response to Reviewer #3]

Response to Comment:

Thank you for the reviewer's comments. We've thoroughly responded to all of other reviewers' comments.

[Additional Modifications]

In addition to the reviewers' comments, we have revised the manuscript to include additional authors due to additional experiments, to emend minor typos, formatting errors, and data entry errors that we discovered during revision. We have also reflected in the revised manuscript all corrected figure and supplementary note numbers and revised references that we did not include in the point-to-point response.

Revised Manuscript (Page 1):

Kyeonghee Lim^{1,3,†}, Jakyoun Lee^{1,3,†}, Sumin Kim^{1,3,†}, Myoungjae Oh^{1,3,†}, Chin Su Koh², Hunkyu Seo^{1,3}, **Yeon-Mi Hong^{1,3}, Won Gi Chung^{1,3}, Jiuk Jang^{1,3}, Hyun Ho Jung^{2*}, Jang-Ung Park^{1,2,3,4*}**

Revised Manuscript (Page 13):

For example, when stimulated with a current density of **0.03 0.075** A mm⁻², the tactile intensity at a finger pressure of 40 kPa was about level 2, but at 100 kPa and 170 kPa it was level 3 and 4, respectively.

Revised Manuscript (Page 34):

Experiment 4: Human-subject perception test controlling pressure

Ten healthy adult subjects (age, 22–35 years; six males and four females) were arranged in the test. Experiment 4 was conducted in the same method of experiment 3 except that the pressure was controlled. The pressure was controlled in 3 conditions (low pressure of 40 kPa, medium pressure of 100 kPa, and high pressure of 170 kPa). The result of the perception test in 3 conditions were compared to the result of the test with no pressure control. Just before subjects put their finger on the device, they cleaned their finger with a pure towel. The pressure from the finger was detected in the dark room using the FET of pressure sensor, where drain voltage of 1 V and gate voltage of 40 V. The voltage was applied by SourceMeter (2400 Series SMU, Keithley), and we monitored the drain current of FET in real-time comparing it to the initial drain current which was recorded before the test. Once a finger of the subject maintained the specified pressure (40 kPa, 100 kPa, and 170 kPa), tactile actuator stimulated his finger in the same method of experiment 3. **The process was repeated five sets (from level 0 to 5) for each subjects.**

Revised Manuscript (Page 35-37):

Experiment 7: Electrotactile perception test with interference stimulation

Ten **healthy adult** subjects (age, 22–35 years; six males and four females) were arranged in the test to test the electrotactile perception test with interference stimulation. **First,** Ag/AgCl electrodes were affixed to the hand and linked to a multi-electrode array for electrophysiological assessment. The signal was recorded using the NeuroPhys/NeuroSorter software (JAGA Systems, Jiang-hi) with a sampling rate of 3060 Hz and a 60 Hz notch filter during the recording process.

Second, ~~e~~Each subject underwent a recognition experiment to ascertain the recognition of the stimulus under single stimulus conditions with varying frequencies. ~~This experiment was repeated 10 times for each subject.~~ Similar procedures were followed for interference stimuli, distinguishing between optimized and non-optimized conditions. This experiment was repeated 10 times for each subject.

Third, ~~t~~The experiment to differentiate between tactile intensity and texture during interference stimulation was designed similarly to experiment 3. **This experiment was repeated once for each subject. Also,** ~~t~~o investigate pressure-dependent changes in tactile intensity, intensity was measured by applying different finger pressures at a current density of 0.0175 A mm⁻² during interference stimulation. The same experiment was conducted under a current density of 0.0225 A mm⁻² for single stimulation. **This experiment was repeated once for each subject.**

Furthermore, to determine the threshold current density required to elicit the "Perceptible" sensation at various pressure levels, the current density was incrementally raised from 0 A mm⁻² to 0.035 A mm⁻² for each type of stimulus (single, interference) until participants reported perceiving the "Perceptible" level of sensation. The current density at that point was then documented.

Last, ~~t~~o discern high-resolution electrotactile patterns, subjects were blindfolded and instructed to place their fingers on the TPIEA, then move their fingers along the direction of the stimulus to learn the pattern. We combined normal stimulation sites achievable through single stimulation, encompassing normal, high-resolution, center, and interference sites resulting from disruptive stimulation. Subsequently, we depicted four distinct patterns—flame, droplet, pinwheel, and leaf. The stimulation was administered sequentially rather than concurrently, enabling the finger to trace the image along the stimulation position from the pattern's inception.

The stimulation frequency (offset frequency) was consistently set at 10 Hz, while the intensity was maintained at a perceptible level. The results were presented in a confusion matrix, and the pattern recognition experiment was repeated five times in a randomized order for each subject.

Revised Manuscript (Page 50-51):

Fig. 5: Electrotactile haptic system with interference stimulation

Revised Manuscript (Page 65):

K. L., J. L., S. K., and M. O. contributed to ideation, conducted the literature research. K. L. participated in the comprehensive planning of the experiment, fabrication of the TPIEA device, execution of tactile perception experiments, participant recruitment, data management, and overall manuscript composition. J. L. was responsible for fabricating and assessing the pressure sensor characteristics and conducting cognitive experiments, while H. S. was involved in measuring finger pressure. M. O. contributed to the overall interference stimulation experiment, and S. K. provided oversight, participated in cognitive experiments, contributed to the design, and offered guidance. Y.-M. H. conducted assessments of skin stability and performed animal testing in safety trials. W. G. C. participated in the measurements of SEP and statistical analysis of data. J. J. provided oversight in the pressure sensor operations. C. S. K. conducted SEP measurements, and H. H. J. supervised the overall SEP

measurements and perception experiments. J.-U. P. supervised, reviewed, and edited the manuscript. All authors participated in the writing and review of this manuscript.

Revised Supplementary Materials (Page 15):

Supplementary Note Figure. 74. Thermal actuation test using electro-tactile actuator.

Revised Supplementary Materials (Page 16):

Supplementary Note Fig. 833. Distribution of the response to the thermal actuation.

(A) Comparison of Gaussian distribution for four conditions of pressure control. (B,C,D,E) Statistical distribution of the response for 10 subjects to the thermal actuation at the uncontrolled pressure (B), at the controlled pressure of 40 kPa (C), 100 kPa (D), and 170 kPa (E).

Revised Supplementary Materials (Page 34):

Supplementary Fig. 1. Four types of mechanoreceptors of sensory touch system on skin. Schematic image showing the four types of mechanoreceptors, the sensory neurons responsible for tactile perception.

Revised Supplementary Materials (Page 43):

Supplementary Fig. 106. Characteristics of pressure-sensitive FET of OF TPIEA. (A) Representative transfer ($V_D = 1$ V) and (B) output ($V_G = 20$ to 70 V) characteristics.

Revised Supplementary Materials (Page 80):

Supplementary Fig. 33. Distribution of the response to the thermal actuation. **(A)** Comparison of Gaussian distribution for four conditions of pressure control. **(B,C,D,E)** Statistical distribution of the response for 10 subjects to the thermal actuation at the uncontrolled pressure **(B)**, at the controlled pressure of 40 kPa **(C)**, 100 kPa **(D)**, and 170 kPa **(E)**.

Revised Manuscript (Page 65):

This work was supported by the Ministry of Science & ICT (MSIT), the Ministry of Trade, Industry and Energy (MOTIE), the Ministry of Health & Welfare, and the Ministry of Food and Drug Safety of Korea through the National Research Foundation (2023R1A2C2006257), ERC Program (2022R1A5A6000846), and the Korea Medical Device Development Fund grant (RMS 2022-11-1209 / KMDF RS-2022-00141392). **This work is funded by Korea Institute of Science and Technology (KIST) Institutional Program (2E33191 and 2E33190).** This work was also supported by Institute for Basic Science (IBS-R026-D1).

Revised Manuscript (Page 65):

K. L., J. L., S. K., and M. O. contributed to ideation, conducted the literature research. K. L. participated in the comprehensive planning of the experiment, fabrication of the TPIEA device, execution of tactile perception experiments, participant recruitment, data management, and overall manuscript composition. J. L. was responsible for fabricating and assessing the pressure sensor characteristics and conducting cognitive experiments, while H. S. was involved in measuring finger pressure. M. O. contributed to the overall interference stimulation experiment, and S. K. provided oversight, participated in cognitive experiments, contributed to the design, and offered guidance. **Y.-M. H. conducted assessments of skin stability and performed animal testing in safety trials. W. G. C. participated in the measurements of SEP and statistical analysis of data. J. J. provided oversight in the pressure sensor operations. C. S. K. conducted SEP measurements, and H. H. J. supervised the overall SEP measurements and perception experiments. J.-U. P. supervised, reviewed, and edited the manuscript. All authors participated in the writing and review of this manuscript.**

REVIEWERS' COMMENTS

Reviewer #1 (Remarks to the Author):

All my questions are addressed. The paper can be accepted.

Reviewer #2 (Remarks to the Author):

Thank you for your detailed response. all issues are addressed very well. This research is very interesting. I believe, once published, it will attract much attentions from the peer. Congrats.

Reviewer #3 (Remarks to the Author):

[Response to Reviewer #1]

Comment : All my questions are addressed. The paper can be accepted.

Response to Comment:

Thank you for the reviewer's comprehensive comments. We appreciate your comments to enhance the manuscript with your insightful comments.

[Response to Reviewer #2]

Comment: Thank you for your detailed response. all issues are addressed very well. This research is very interesting. I believe, once published, it will attract much attentions from the peer. Congrats.

Response to Comment:

Thank you for the reviewer's positive comments. We are thankful for the opportunity to enhance the quality of our manuscript.

[Response to Reviewer #3]

Comment: I co-reviewed this manuscript with one of the reviewers who provided the listed reports. This is part of the Nature Communications initiative to facilitate training in peer review and to provide appropriate recognition for Early Career Researchers who co-review manuscripts.

Response to Comment:

Thank you for the reviewer's comments. We've thoroughly responded to all of other reviewers' comments.